# MULTI-EPOCH MATRIX FACTORIZATION MECHANISMS FOR PRIVATE MACHINE LEARNING

## ABSTRACT

We introduce new differentially private (DP) mechanisms for gradient-based machine learning (ML) training involving multiple passes (epochs) of a dataset, substantially improving the achievable privacy-utility-computation tradeoffs. Our key contribution is an extension of the online matrix factorization DP mechanism to multiple participations, substantially generalizing the approach of Denisov et al. (2022). We first give conditions under which it is possible to reduce the problem with per-iteration vector contributions to the simpler one of scalar contributions. Using this, we formulate the construction of optimal (in total squared error at each iterate) matrix mechanisms for SGD variants as a convex program. We propose an efficient optimization algorithm via a closed form solution to the dual function.

While tractable, both solving the convex problem offline and computing the necessary noise masks during training can become prohibitively expensive when many training steps are necessary. To address this, we design a Fourier-transform-based mechanism with significantly less computation and only a minor utility decrease.

Extensive empirical evaluation on two tasks: example-level DP for image classification and user-level DP for language modeling, demonstrate substantial improvements over the previous state-of-the-art. Though our primary application is to ML, we note our main DP results are applicable to arbitrary linear queries and hence may have much broader applicability.

## 1 INTRODUCTION

Differentially private stochastic gradient descent (DP-SGD) is the de facto standard algorithm for DP machine learning (ML) (Song et al., 2013; Abadi et al., 2016a). However, obtaining state-of-the-art privacy-utility tradeoffs critically requires use of privacy amplification techniques like shuffling (Erlingsson et al., 2019; Feldman et al., 2022) or (Poisson) subsampling (Bassily et al., 2014; Zhu & Wang, 2019; Wang et al., 2019). These in turn require strong assumptions on the manner in which data is processed that are rarely valid in applications of DP-SGD, as implementing these procedures is often impractical (Kairouz et al., 2021).

Kairouz et al. (2021) recently proposed the DP-FTRL framework that avoids reliance on amplification by sampling, through using DP streaming of prefix sums (Dwork et al., 2010; Chan et al., 2011; Honaker, 2015). DP-FTRL can often match (or outperform) DP-SGD in privacy-utility tradeoffs. Indeed, this algorithm enabled McMahan & Thakurta (2022) to train the first known provably DP ML model on user data in a production setting.

Several works have since focused on this primitive as an instantiation of the streaming matrix mechanism; in particular, Denisov et al. (2022) showed that leveraging optimal matrix mechanisms led to significant empirical improvements, though their work was restricted to the single-epoch setting. Shown in Figs. 1 and 3, we achieve substantially improved privacy-utility tradeoffs, with comparable computation. Our methods outperform all prior work, including DP-SGD with amplification, to as low as $\varepsilon \approx 2$. To accomplish this, we propose a formalism for measuring *multi-participation sensitivity*, given in Section 2, a significant extension to the single-participation sensitivity used in in Denisov et al. (2022). We show in Section 3 how one may compute matrix mechanisms optimized for this multi-participation setting. This generalization enables application of optimized streaming matrix mechanisms to settings where each example (or user) may contribute to multiple elements of the data matrix (the matrix formed by stacking unnoised batch gradients in ML).

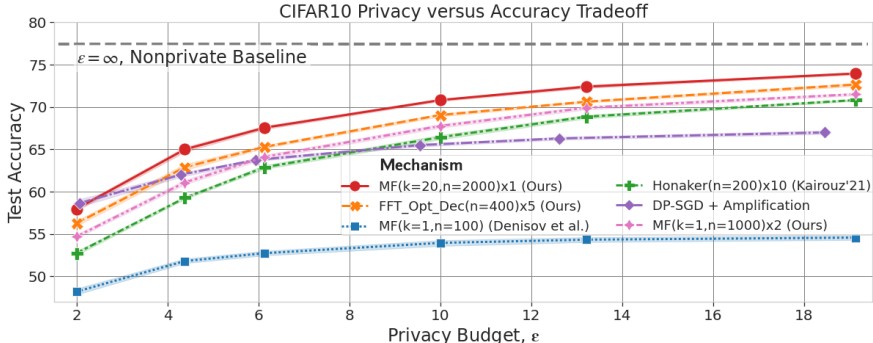

Figure 1: **Our optimal multi-epoch matrix and FFT-based mechanisms outperform all others, including DP-SGD with amplification**, as low as $\varepsilon \approx 4$. Using our sensitivity calculation of Theorem 2.1 and stamping (Section 5), we optimize a single pass ($k = 1$) matrix of Denisov et al. (2022) but apply it here with $> 1$ pass. We use an online Honaker-based decoder equivalent to that of Kairouz et al. (2021) except for a significant improvement to tree-completion in Appendix D.3. Models trained for 20 epochs on CIFAR10 with a batch size of 500. We repeat each setting 12 times and show 95% bootstrapped confidence intervals. Empirical setup is in Section 5.1.

We also explore the computational tradeoffs of our approaches. In particular, computing optimal matrix factorizations may become intractable when large numbers of training steps $n$ are required , as we discuss in Section 4. While this is uncommon in the federated algorithms for user-level DP, it can be a limitation when training with SGD for example-level privacy. To reduce this cost, we propose and investigate an approach based on the Fast Fourier Transform (FFT) (Nussbaumer, 1981), which is near-optimal for the single-epoch setting and efficiently computable for most, if not all, ML settings. Indeed, we find this approach still outperforms the mechanisms from the extant literature, even under multiple participations.

**Contributions** 1) We provide a framework for computing the sensitivity of matrix mechanisms under general participation schemas. To do this, we prove a new theorem bounding sensitivity for multi-dimensional data contributions. This allows us to reduce the problem to that of measuring sensitivity for scalar contributions alone (Section 2). 2) We extend the results of Denisov et al. (2022) to the optimization problems corresponding to these generalized notions of sensitivity, showing that the algorithms proposed there can be applied in our setting (Section 3). 3) We propose and analyze a computationally-efficient factorization based on the Fourier transform which is near optimal for the single-epoch setting and can be efficiently extended to handle multiple epochs (Section 4). 4) We perform detailed empirical comparisons of our mechanisms with both the prior matrix mechanism approaches and DP-SGD. We show that the methods proposed here outperform all others (in particular, DP-SGD with amplification), to privacy budgets as low as $\varepsilon \approx 2$, and without any need for privacy amplification (Section 5). 5) We will upload all code used in the final manuscript.

**Related work** The core privacy primitive here is the matrix mechanism (Li et al., 2015). Its long history of study and application was mostly in the offline setting (McKenna et al., 2018; Edmonds et al., 2020; Yuan et al., 2016; Hardt & Talwar, 2010). Fichtenberger et al. (2022); Denisov et al. (2022) independently applied it to the *adaptive streaming* setting, where outputs are released one-by-one and privacy analysis must account for an adversary adaptively defining the inputs. Denisov et al. (2022) connected the matrix mechanism to DP ML, via the DP-FTRL algorithm of Kairouz et al. (2021), and showed that computing optimal factorizations significantly improves the privacy-utility-computation tradeoffs when needing only a single pass (epoch) over the training data.

**Example- and user-level DP, and the connection to federated learning (FL)** In addition to example-level DP, we consider user-level DP. As observed by McMahan et al. (2018), private FL algorithms are well suited to providing user-level DP or other multi-example units of privacy, e.g. document-level, as bounding the sensitivity of a single user's contribution to an aggregate update is made straightforward by the per-user data processing pattern inherent in FL. However, our primary application is to datacenter training, where user data can be processed in a fixed shuffled order, unlike cross-device FL. We use the term 'participation' to denote the event that an example (user or client in FL) contributes to the gradient sum (or a model update in FL) $\mathbf{x}_i$ for a given step/iteration (round in FL) $i$. Individual contributions to $\mathbf{x}_i$ are scaled so their maximum $\ell_2$ norm is $\zeta$. Our

mechanisms compute sums over individual clipped contributions, and then post-process by dividing by the batch size (or clients/round) to compute an average gradient (or model update). We assume $\zeta = 1$, applying appropriate scaling as needed. Appendix A summarizes terminology and notation.

## 2    PRIVACY FOR ADAPTIVE STREAMS WITH MULTIPLE PARTICIPATIONS

We define and efficiently bound the sensitivity of the multi-participation adaptive streaming (continual release) setting, by generalizing Denisov et al. (2022, Sec. 2). We assume a database of $m$ examples (or users in FL, or records in a general DP application) that is processed as a stream over $n$ steps. A set of $B$ examples is selected on each step $i$, and processed via an adaptively chosen function (e.g., computing a gradient at the current model), producing a vector of $\ell_2$ norm at most $\zeta$. These vectors are summed and provided to the DP mechanism as $\mathbf{x}_i \in \mathbb{R}^d$, which then releases a privatized function of $[\mathbf{x}_1, \dots, \mathbf{x}_i]$, the stream so far. When a particular example contributes to the sum $\mathbf{x}_i$, we say it *participates* on step $i$. We are primarily interested in the case where $m/B < n$, and hence each example is used on more than one step. This is the multiple epoch setting of ML.

Two data streams $\mathbf{x}$ and $\tilde{\mathbf{x}}$ are said to be neighboring if they differ in the contributions derived from a single example, either by zeroing out all of its contributions, or by replacing them arbitrarily subject to the norm bound $\zeta$. Thus, the participation pattern does not change (all records contribute to the same steps in $\mathbf{x}$ and $\tilde{\mathbf{x}}$, with only the vectors associated with one record changing). We define a *participation schema* $\Pi$ as the set of possible *participation patterns* $\pi \in \Pi$, with each $\pi \subseteq [n]$ indicating a set of steps in which a single example might participate. Assuming each record contributes at most once (*single-participation*, $\Pi = \{\{1\}, \{2\}, \dots \{n\}\}$), recovers the standard streaming setting. This captures, for example, training with minibatch SGD using a single pass (epoch) over a training dataset. At the other extreme, we have *every-step participation* with $\Pi = \{[n]\}$ where each record contributes to every step. This captures learning with full gradient descent, where we compute the gradient on the full training dataset on every iteration.

**Fixed-epoch-order participation**    We focus on generalization of the above two, $(k, b)$-*participation*, where each example participates at most $k$ times, with any adjacent participations exactly $b$ steps apart: formally, $\Pi$ is the set of all $\pi$ such that $|\pi| \le k$, and if $\pi = \{i_1, \dots, i_k\}$, we have $\forall j \in \{2, \dots, k\}, i_j - i_{j-1} = b$. Note $(k{=}1, b{=}n)$-participation recovers the single-epoch setting, and $(k{=}n, b{=}1)$-participation recovers every-step participation, and for example $(k{=}3, b{=}2)$-participation has $\Pi = \{\{1, 3, 5\}, \{2, 4, 6\}\}$. We focus on this participation schema because: 1) It encompasses multi-epoch SGD training using a data processing pattern well-supported by modern ML infrastructure.[1] The only requirement is that rather than shuffling the dataset for each epoch, the dataset is shuffled once and the same order of minibatches is used for each epoch. With this setup, $k$ epochs of training on a dataset of size $m$ with a batch size $B$ gives $n = mk/B$ total training steps, and satisfies $(k, m/B)$-participation. 2) We show (e.g., Eq. (3)) in importance cases this participation schema allows for the efficient computation of sensitivity. 3) We will see in Section 3 that the more possible participation patterns $\pi$, the more constrained the problem of finding optimal mechanisms becomes. Hence, a relatively restrictive (but practical) schema like $(k, b)$-participation yields more favorable privacy-utility tradeoffs.

**Sensitivity of linear queries on multi-participation adaptive streams**    Consider a full-rank square query (or workload) matrix $\mathbf{A} \in \mathbb{R}^{n \times n}$; we wish to compute the function $\mathbf{x} \mapsto \mathbf{Ax}$ in a differentially private manner, where we consider inputs $\mathbf{x}$ and outputs $\mathbf{Ax}$ to be elements of $\mathbb{R}^{n \times d}$, under geometry inherited from the Frobenius inner product. We utilize the matrix mechanism (Li et al., 2015), which, provided a factorization $\mathbf{A} = \mathbf{BC}$, computes the estimate

$$\widehat{\mathbf{Ax}} = \mathbf{B}\left(\mathbf{Cx} + \mathbf{z}\right), \tag{1}$$

where $\mathbf{z}$ is a sample from appropriately scaled isotropic Gaussian noise. The scale is determined by the *sensitivity* of the mapping $\mathbf{x} \mapsto \mathbf{Cx}$; roughly, how much outputs of this mapping can vary (in $\ell_2$

---

[1]This is in contrast to Poisson, or independent fixed-sized batch sampling, with replacement across steps, as is assumed by many works (Abadi et al., 2016a; Bassily et al., 2014; Zhu & Wang, 2019; Wang et al., 2019). Many works in fact process batches in a shuffled order without replacement and then incorrectly apply DP analysis for, e.g., Poisson sampling. Indeed, we perform this same—incorrect—analysis for our DP-SGD baseline because it reproduces the previous state-of-the-art results for DP-SGD.

norm) when we swap the input stream $\mathbf{x}$ for a neighboring one $\tilde{\mathbf{x}}$. We refer to this matrix $\mathbf{C}$ as the "encoder", as it encodes $\mathbf{x}$ as $\mathbf{Cx}$ before adding Gaussian noise. Similarly, we call $\mathbf{B}$ the "decoder".

Let $\mathbf{N}$ be the set of all pairs of neighboring streams $\mathbf{x}$ and $\mathfrak{D} := \{\mathbf{x} - \tilde{\mathbf{x}} \mid (\mathbf{x}, \tilde{\mathbf{x}}) \in \mathbf{N}\}$ represent the set of all possible deltas between neighboring $\mathbf{x}, \tilde{\mathbf{x}}$. The definition of $\mathfrak{D}$ implies it is symmetric ($\mathbf{u} \in \mathfrak{D} \Rightarrow -\mathbf{u} \in \mathfrak{D}$). We will say a $\mathfrak{D}$ **satisfies the participation schema** $\Pi$ if the indices of all nonzero elements in each vector $\mathbf{u} \in \mathfrak{D}$ corresponds some $\pi \in \Pi$. Critically, for linear queries $\mathfrak{D}$ fully captures the sensitivity of the query:

**Definition 1.** *The **sensitivity** of the matrix factorization mechanism Eq.* (1) *is defined as*

$$\text{sens}_{\mathfrak{D}}(\mathbf{C}) = \sup_{(\mathbf{x}, \tilde{\mathbf{x}}) \in \mathbf{N}} \|\mathbf{Cx} - \mathbf{C}\tilde{\mathbf{x}}\|_F = \sup_{\mathbf{u} \in \mathfrak{D}} \|\mathbf{Cu}\|_F. \tag{2}$$

Convexity of $\|\mathbf{Cu}\|_F$ in $\mathbf{u}$ implies that $\sup_{\mathbf{u} \in \mathfrak{D}} \|\mathbf{Cu}\|_F = \sup_{\mathbf{u} \in \text{conv}(\mathfrak{D})} \|\mathbf{Cu}\|_F$, and hence without loss of generality (wlog), we take $\mathfrak{D}$ to be convex as needed. It is illustrative to consider some specific $\mathfrak{D}$s for scalar per-step contributions with $\zeta = d = 1$. Single-participation corresponds to $\mathfrak{D} = \text{conv}\{\alpha e_i | \alpha \in [-1, 1], i \in [n]\}$ where $e_i$ for $i \in [n]$ are the standard basis vectors. Noting $\|\mathbf{Cu}\| = \| - \mathbf{Cu}\|$ and convexity of $\|\mathbf{Cu}\|$, we see the maximum will be achieved at some $e_i$, recovering the 'max-$\ell_2$-norm-over-columns' measurement of sensitivity of Li et al. (2015, Proposition 3). Every-step participation corresponds to the $\ell_\infty$ ball, $\mathfrak{D} = \{\mathbf{x} \mid \|\mathbf{x}\|_\infty \le 1\}$.

**Conditions allowing the reduction to per-iterate scalar contributions** In ML, examples are used to calculate gradients of $d > 1$ dimensions, and so we wish to consider $\mathbf{x} \in \mathbb{R}^{n \times d}$, with rows $\mathbf{x}_i \in \mathbb{R}^d$ corresponding to the sum of gradients for examples participating in step $i$. In order to compute sensitivity, one may hope that the sensitivity for each $\mathbf{x}_i \in \mathbb{R}^d$ can be bounded by only considering some appropriately worst-case $x_i \in \mathbb{R}$. More formally, consider a fixed participation schema $\Pi$, and further assume (wlog) $\zeta = 1$. Then, for vector-valued contributions we have

$$\mathfrak{D}_\Pi^d = \text{conv}\left\{ \mathbf{G} \in \mathbb{R}^{n \times d} \mid \exists \pi \in \Pi \text{ s.t. } \|\mathbf{G}_{[i,:]}\|_2 \le 1 \text{ for } i \in \pi \text{ and } \mathbf{G}_{[i,:]} = \mathbf{0} \text{ for } i \notin \pi \right\}.$$

In the $d = 1$ case, we have a much simpler polytope, $\mathfrak{D}_\Pi^1 = \text{conv}(\mathcal{D}_\Pi^1)$ where

$$\mathcal{D}_\Pi^1 = \bigcup_{\pi \in \Pi} \left\{ \mathbf{u} \in \mathbb{R}^n \mid \mathbf{u}_i \in \{-1, 1\} \text{ if } i \in \pi, 0 \text{ otherwise} \right\}.$$

One might hope to show $\text{sens}_{\mathfrak{D}_\Pi^d}(\mathbf{C}) \le \text{sens}_{\mathfrak{D}_\Pi^1}(\mathbf{C})$, and the authors in fact initially conjectured this to be true. To our surprise, while this inequality holds under a variety of assumptions, it does not hold in general (Appendix H.2 gives a counterexample).[2] Empirically we have observed that for various query matrices $\mathbf{A}$ and $(k, b)$-participation with $d = 1$, the optimal $\mathbf{C}$ satisfy (or almost satisfy) the condition $\mathbf{C}^\top \mathbf{C} \ge 0$ (element-wise non-negativity). In this case, we can show:

**Corollary 2.1.** *When per-step contributions bounded by $\zeta = 1$, for any participation schema $\Pi$ and dimensionality $d \ge 1$, when $\mathbf{C}^\top \mathbf{C} \ge 0$ elementwise, we have $\text{sens}_{\mathfrak{D}_\Pi^d}(\mathbf{C}) = \text{sens}_{\mathfrak{D}_\Pi^1}(\mathbf{C})$.*

In particular, this implies that if $\mathbf{C}$ is optimal in the $d = 1$ case and satisfies $\mathbf{C}^\top \mathbf{C} \ge 0$, it is also optimal in the $d > 1$ case. This result is a corollary of Theorem H.1, which establishes additional conditions under which $\text{sens}_{\mathfrak{D}_\Pi^d}(\mathbf{C}) \le \text{sens}_{\mathfrak{D}_\Pi^1}(\mathbf{C})$ holds. All proofs are in Appendix H onwards.

**Difficulty of computing** $\text{sens}(\mathbf{C})$ In general, computing $\text{sens}(\mathbf{C})$ is a convex quadratic *maximization* problem over a convex set, which can be NP-hard. Even the simple case of computing the sensitivity for an arbitrary matrix $\mathbf{C}$ under every-step participation with scalar ($d = 1$) contributions is NP-hard—it is exactly the problem of computing the $\ell_\infty - \ell_2$ operator norm (Tropp, 2004). (In fact, it is useful to observe $\text{sens}_{\mathfrak{D}}(\cdot)$ can always be viewed as an operator norm, see Appendix B). This hardness is in stark contrast to the single-participation setting, where calculating sensitivity is trivial. However, we *can in some cases compute sensitivity exactly by* **brute force**. Take $d = 1$. Observe $\mathcal{D}_\Pi^1$ is a finite set and so a direct calculation by using Eq. (2) is often possible. But, $|\mathcal{D}_\Pi^1| = |\Pi|2^k$, and observing the symmetry $\|\mathbf{Cu}\| = \|\mathbf{C}(-\mathbf{u})\|$ can reduce the computational cost only by half. In general $|\Pi|$ may be exponential in $k$, but in the special case of $(k, b)$-participation, we have $|\Pi| = b$

---

[2]We conjecture it is "almost" true; tightly bounding the necessary error term is an interesting open question.

(the number of steps in one epoch). Hence, for modest numbers of epochs $k$, directly computing sensitivity is possible, e.g., in our StackOverflow experiments in Section 5.2, we can reduce $\mathbf{u} \in \mathcal{D}_{\Pi}^1$ to only $342 \cdot 2^5 = 10,944$ vectors. Theorem H.1 can be used to translate bounds from scalar to higher dimensions, though without such a translation it is not clear how to generalize a brute-force method.

**Computing sensitivity when $\mathbf{C}^\top \mathbf{C} \geq 0$**   Let $\mathbf{X} = \mathbf{C}^\top \mathbf{C}$. When $\mathbf{X}$ has only nonnegative elements, one may reduce the problem of computing $\mathrm{sens}_{\mathfrak{D}_{\Pi}^1}(\mathbf{C})$ to

$$\mathrm{sens}_{\mathfrak{D}_{\Pi}^d}(\mathbf{C}) = \max_{\mathbf{u} \in \mathfrak{D}_{\Pi}^1} \|\mathbf{C}\mathbf{u}\|_F = \max_{\mathbf{u} \in \mathfrak{D}_{\Pi}^1} \sqrt{\mathbf{u}^\top \mathbf{X}\mathbf{u}} = \max_{\pi \in \Pi} \sqrt{\mathbf{1}^\top \mathbf{X}_{[\pi,\pi]}\mathbf{1}}, \tag{3}$$

where $\mathbf{X}_{[\pi,\pi]} \in \mathbb{R}^{k \times k}$ is the submatrix of $\mathbf{X}$ formed from the rows and columns selected by $\pi$, $|\pi| = k$ and $\mathbf{1} \in \mathbb{R}^k$. The first equality follows from Corollary 2.1, and then the max must be achieved by the maximum-magnitude nonnegative vector $\mathbf{u}$, specifically $\mathbf{1}^k$. As noted above, the matrices we consider satisfy this property, and hence we can compute the exact sensitivity efficiently for $(k, b)$-participation.

**Upper-bounding sensitivity**   As an alternative to structural conditions on $\mathbf{C}$ or $\mathbf{X}$ allowing efficient exact computation of sensitivity for $d > 1$, we can look to (reasonably tight) upper bounds on the sensitivity of $\mathbf{C}$. In the case of $(k, b)$-participation, one efficient method of computing upper bounds for the multiple-participation sensitivity of $\mathbf{C}$ has shown itself to be particularly useful:

**Theorem 2.1.** *Let $\mathbf{C} \in \mathbb{R}^{n \times n}$, and take some participation schema $\Pi$, with $k = \max_{\pi \in \Pi} |\pi|$ the maximum number of participations. With $\mathbf{C}_{[:,\pi]}$ representing selecting the columns of the matrix $\mathbf{C}$ indexed by $\pi$ and $\|\cdot\|_2$ the spectral matrix norm, let $\lambda = \max_{\pi \in \Pi} \left\|\mathbf{C}_{[:,\pi]}\right\|_2$. Then $\mathrm{sens}_{\mathfrak{D}_{\Pi}^1}(\mathbf{C}) \leq \lambda\sqrt{k}$.*

In the $(k, b)$-participation case, $|\Pi| = b$. The complexity of computing the largest eigenvalue of the subselected $\mathbf{C}$ matrix is cubic in $k$. Thus, computing this upper bound is of order $bk^3$, easily computable for the range of $k, b$ considered here ($k \leq 100, b \leq 500$).

**Differential Privacy Guarantee**   Using our generalization of adaptive streams to multiple participations we obtain the following result (a straightforward generalization of Denisov et al. (2022, Theorem 2.1)). The proof is identical that in Denisov et al. (2022), except we replace the sensitivity bound with that for multiple participations obtained via Corollary 2.1.

**Theorem 2.2.** *Let $\mathbf{A} \in \mathbb{R}^{n \times n}$ be a lower-triangular full-rank query matrix, and let $\mathbf{A} = \mathbf{B}\mathbf{C}$ be any factorization with the following property: for any two neighboring streams $\mathbf{x}, \tilde{\mathbf{x}} \in \mathbb{R}^{n \times d}$, we have $\|\mathbf{C}(\mathbf{x} - \tilde{\mathbf{x}})\|_F \leq \kappa$. Let $\mathbf{Z} \sim \mathcal{N}(0, \kappa^2\sigma^2)^{n \times d}$ with $\sigma$ large enough so that $\mathcal{M}(\mathbf{x}) = \mathbf{A}\mathbf{x} + \mathbf{B}\mathbf{Z} = \mathbf{B}(\mathbf{C}\mathbf{x} + \mathbf{Z})$ satisfies $(\varepsilon, \delta)$-DP (or $\rho$-zCDP or $\mu$-Gaussian DP) in the nonadaptive continual release model. Then, $\mathcal{M}$ satisfies the same DP guarantee (with the same parameters) even when the rows of the input are chosen adaptively.*

## 3   OPTIMIZING MATRIX MECHANISMS FOR MULTIPLE EPOCHS

We now present methods for computing optimal matrix mechanisms that are specialized to a specific participation schema $\Pi$ and query matrix $\mathbf{A}$. For example, Fig. 2 shows the optimal factorization for the query matrix $\mathbf{A}$ representing SGD with momentum and coldown under ($k{=}6, b{=}342$)-participation, used in Section 5.2. Specializing the mechanism to both the participation pattern and the specific query workload enables as to obtain state-of-the-art results in ML (Section 5).

We follow the approach of Denisov et al. (2022) which showed empirical improvements over prior methods in single-epoch settings. We begin by defining the loss of interest, i.e., the total variance of noise added, for the mechanism defined in Eq. (1). Note that this loss characterizes other downstream tasks like DP Mean Estimation. Given $\mathfrak{D}$, assume that we may represent $\mathfrak{D} = \mathrm{conv}(\mathcal{D})$ for some finite set $\mathcal{D}$—as we have seen, this is the case, e.g., in $(k, b)$-participation. Then the loss which corresponds to total squared error of a factorization, at a fixed privacy level, may be expressed as:

$$\mathcal{L}(\mathbf{B}, \mathbf{C}) = \mathrm{sens}_{\mathfrak{D}}^2(\mathbf{C}) \|\mathbf{B}\|_F^2 \qquad \text{where} \qquad \mathrm{sens}_{\mathfrak{D}}(\mathbf{C}) = \sup_{\mathbf{u} \in \mathfrak{D}} \|\mathbf{C}\mathbf{u}\|_2^2 = \sup_{\mathbf{u} \in \mathcal{D}} \|\mathbf{C}\mathbf{u}\|_2^2. \tag{4}$$

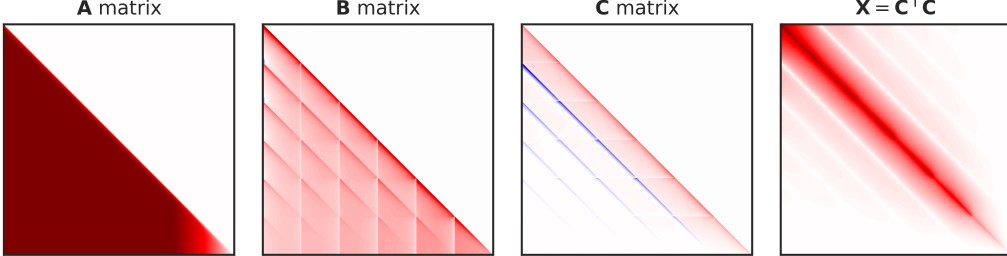

Figure 2: The optimal factorization $\mathbf{A} = \mathbf{BC}$ under $(k=6, b=342)$-participation, constructed by solving the optimization problem Eq. (5). Matrix $\mathbf{A}$ encodes SGD with momentum 0.95 and a learning-rate cooldown schedule for the last 25% of rounds, as used in our StackOverflow experiments (Section 5.2). The constraints on sensitivity imposed by this participation schema are evident in the resulting matrices. For example, the white diagonals with a period of $b = 342$ in $\mathbf{X} = \mathbf{C}^\top \mathbf{C}$ show that the columns of $\mathbf{C}$ that could correspond to a pair of rounds $(i, j)$ where the same user might participate are in fact orthogonal. See Fig. 13 in Appendix F.4 for a larger view.

Observing that $\forall \alpha$ the mechanism $\mathbf{A} = (\alpha\mathbf{B})(\frac{1}{\alpha}\mathbf{C})$ has identical loss, we conclude that we may consider the constrained version of the problem of minimizing this loss where $\mathrm{sens}_{\mathfrak{D}}(\mathbf{C}) \leq 1$. Since for any $\mathbf{C}$, $\mathbf{B} = \mathbf{AC}^\dagger$ produces the minimum-Frobeneous norm $\mathbf{B}$-matrix, it is sufficient to solve:

$$\min_{\mathbf{C}} \mathcal{L}\left(\mathbf{AC}^\dagger, \mathbf{C}\right) = \min_{\mathbf{C}:\mathrm{sens}_{\mathfrak{D}}^2(\mathbf{C})=1} \left\|\mathbf{AC}^\dagger\right\|_F^2. \tag{5}$$

With the change of variables $\mathbf{X} = \mathbf{C}^\top\mathbf{C}$, equivalently:

$$\min_{\mathbf{X}\text{ is PD}, \widehat{\mathrm{sens}}_{\mathfrak{D}}^2(\mathbf{X})=1} \mathrm{tr}(\mathbf{A}^\top\mathbf{A}\mathbf{X}^{-1}) \quad \text{where} \quad \widehat{\mathrm{sens}}_{\mathfrak{D}}^2(\mathbf{X}) \leq \sup_{\mathbf{u}\in\mathcal{D}} \mathbf{u}^\top\mathbf{X}\mathbf{u}. \tag{6}$$

**Theorem 3.1.** *Let a finite $\mathcal{D} = \{\mathbf{u}_i\}_{i=1}^k$ be given, and assume that the vectors $\{\mathbf{u}_i\}_{i=1}^k$ span $\mathbb{R}^n$. Assume that $\mathbf{A}$ is full-rank, and for $\mathbf{v} \in \mathbb{R}^k$ define $\mathbf{H}_\mathbf{v} = [\mathbf{u}_1, \ldots, \mathbf{u}_k]\,\mathrm{diag}(\mathbf{v})^{1/2}$, $\mathbf{U} = \mathbf{H}_\mathbf{v}\mathbf{H}_\mathbf{v}^\top$. Define the Lagrangian $L$ as $L(\mathbf{X}, \mathbf{v}) := \mathrm{tr}(\mathbf{A}^\top\mathbf{A}\mathbf{X}^{-1}) + \sum_{\mathbf{u}\in\mathcal{D}} \mathbf{v}_\mathbf{u}\left(\mathbf{u}^\top\mathbf{X}\mathbf{u} - 1\right)$. Then, for Lagrange multipliers $\mathbf{v}$ such that the $\mathbf{U}$ is full-rank, the minimizer $\mathbf{X}(\mathbf{v})$ of $L$ for this fixed $\mathbf{v}$ may be represented $\mathbf{X}(\mathbf{v}) = \mathbf{U}^{-\frac{1}{2}}\left(\mathbf{U}^{\frac{1}{2}}\mathbf{A}^\top\mathbf{A}\mathbf{U}^{\frac{1}{2}}\right)^{\frac{1}{2}}\mathbf{U}^{-\frac{1}{2}}$. and the Lagrange dual function $g$ for the problem Eq. (6) can be expressed in closed form in terms of the dual variables $\mathbf{v}$:*

$$g(\mathbf{v}) := \inf_{\mathbf{X}\text{ is PD}} L(\mathbf{X}, \mathbf{v}) = 2\,\mathrm{tr}\left(\left(\mathbf{U}^{\frac{1}{2}}\mathbf{A}^\top\mathbf{A}\mathbf{U}^{\frac{1}{2}}\right)^{\frac{1}{2}}\right) - \sum_{\mathbf{u}\in\mathcal{D}} \mathbf{v}_\mathbf{u} \tag{7}$$

**Remark.** The restriction that $\mathbf{v}$ yields a full-rank $\mathbf{U}$ serves to restrict to cases where the Lagrangian has a finite, positive-definite minimizer in the primal variable; if the vectors $\{\mathbf{u}\}$ span $\mathbb{R}^n$, the problem Eq. (6) has a finite minimizer by Lemma I.1. Any setting of the dual variables $\mathbf{v}$ corresponding to this minimizer is contained in a neighborhood uniformly satisfying this full-rank property, and so it is valid to differentiate our expression for $g$ with respect to such $\mathbf{v}$ (as we will do in Appendix I.1).

**Corollary 3.1.** *In the same setup as Theorem 3.1, the gradient of the dual function $g$ is:* $\frac{\partial g}{\partial \mathbf{v}_i} = \mathbf{u}_i^\top\mathbf{U}^{-\frac{1}{2}}\left(\mathbf{U}^{\frac{1}{2}}\mathbf{A}^\top\mathbf{A}\mathbf{U}^{\frac{1}{2}}\right)^{\frac{1}{2}}\mathbf{U}^{-\frac{1}{2}}\mathbf{u}_i - 1$. *Moreover, a maximizer of the dual $\mathbf{v}^\star$ must satisfy:*

$$\mathbf{v}^\star = \mathrm{diagpart}\left(\left(\mathbf{H}_{\mathbf{v}^\star}^\top\mathbf{A}^\top\mathbf{A}\mathbf{H}_{\mathbf{v}^\star}\right)^{\frac{1}{2}}\right). \tag{8}$$

*The optimal value of the problem defined in Eq. (6) is $\mathrm{tr}(\mathbf{v}^\star)$.*

**Remark.** In the single-participation case of Denisov et al. (2022), $\mathbf{H}_\mathbf{v} = \mathrm{diag}(\mathbf{v})^{\frac{1}{2}}$, and Eq. (8) recovers the fixed point expression of that paper's Theorem 3.2. Our Corollary 3.1 implies that the optimization methods presented in Denisov et al. (2022) may be applied, with suitable translation, to our setting; we use these methods to generate the optimal matrices studied empirically in Section 5.

## 4 FAST-FOURIER-TRANSFORM-BASED DP-PREFIX SUM ESTIMATION

Our work has two types of computation costs: **optimization costs** are those associated with optimizing and generating (or, computing) a mechanism whereas **noise generation costs** are those

associated with using the mechanism to sample noise for DP prefix sum release. Note that once optimized, a single mechanism can be reused indefinitely to generate noise for other runs by simply resampling new noise and applying the same decoder. The best known methods for computing the optimal factorizations scale as at least $\mathcal{O}(n^3)$ (Yuan et al., 2016; Denisov et al., 2022). This optimization cost can become intractable when $n$ grows too large. Thus, in this section we focus on reducing optimization computation at a small decrease in the achievable privacy-utility tradeoff.

A prime candidate for this goal is the Discrete Fourier Transform (DFT) because there are known algorithms both for nearly-optimal private convolutions (Fawaz et al., 2013) which are intimately related to the DFT, and for efficient calculation of the DFT using the Fast Fourier Transform (FFT) (Nussbaumer, 1981). We present an FFT-based mechanism that reduces the noise generation costs. We then prove rigorous DP guarantees for it and show that these lead to near-optimal privacy-utility tradeoffs in the single-epoch setting. We provide two improvements over prior work: 1) extending the result to the multi-epoch and multi-dimensional setting and 2) providing explicit non-asymptotic analysis of the algorithm's utility.

Let $\mathbf{A}$ represent the (Toeplitz) matrix of all 1s on or below the main diagonal and 0s elsewhere; i.e., the prefix-sum matrix. In this section, we perform our analysis in the Fourier domain. To release $\mathbf{A}\mathbf{x}$, we define a circulant matrix $\mathbf{A}_{\text{circ}} \in \mathbb{R}^{2n \times 2n}$ with a corresponding input vector $\mathbf{x}_{\text{ext}} = \text{concat}(\mathbf{x}, \mathbf{0}_n), \mathbf{0}_n \in \{0\}^n$ so that the first $n$ entries of $\mathbf{A}_{\text{circ}}\mathbf{x}_{\text{ext}}$ are equal to $\mathbf{A}\mathbf{x}$ (see Appendix J.1). Thus, we study the DP release of $\mathbf{A}_{\text{circ}}\mathbf{x}_{\text{ext}}$. Note, to be consistent with the literature on FFT, in this section, and in Appendix J, we will index all the vectors and matrices with starting index of *zero*.

Theorem J.1 of Gray (2006) (restated in Appendix J.1) shows there exists a diagonal $\Sigma$ such that $\mathbf{A}_{\text{circ}} = \mathbf{F}^* \Sigma \mathbf{F}$ for diagonal $\Sigma$, where $\mathbf{F}$ is the DFT matrix. Then, $\mathbf{A}_{\text{circ}}$ can then be factorized as $\mathbf{A}_{\text{circ}} = \mathbf{B}_{\text{circ}}\mathbf{C}_{\text{circ}}$ where $\mathbf{B}_{\text{circ}} = \mathbf{F}^* \Sigma^{1/2}$ and $\mathbf{C}_{\text{circ}} = \Sigma^{1/2}\mathbf{F}$.

The (complex-valued) matrix mechanism specified by the factorization above (and presented as Algorithm 1 of Appendix C) is nearly optimal in the class of matrix-factorization-based mechanisms, as we show in Appendix J. Though we prove a simple zCDP guarantee for *any* participation schema having at most $k = \max_{\pi \in \Pi} |\pi|$ participations in Theorem 4.1, we can instead use our Corollary 2.1 when the participation schema is known in advance (as in our experiments with $(k, b)$-participation).

**Theorem 4.1.** *Under $k$-participation, Algorithm 1 satisfies $(k^2\rho)$-zCDP.*

**The optimal FFT decoder** Observe from Theorems 2.1 and 2.2 that the privacy guarantee of our multi-participation adaptive setting is independent of the choice of decoder, $\mathbf{B}$. Thus, instead of taking $\mathbf{B}_{\text{circ}}$ above, we take the optimal decoder using the Moore-Penrose pseudoinverse of the encoder, i.e., $\mathbf{B}_{\text{circ}} = \mathbf{S}\mathbf{C}_{\text{circ}}^{\dagger}$. We do so using an equivalent encoder $\mathbf{C} \in \mathbb{R}^{n \times 2n}$ as shown in Appendix K. We find that this leads to significant improvements in the privacy-utility tradeoff (see Fig. 9 in Appendix E) at no additional computational overhead.

**Computation Costs** Though we define the optimal FFT decoder as a pseudoinverse, observe that we do not need to optimize (or even compute) the decoder; by Appendix K, the problem is reduced to that of solving a highly structured linear system. However, we find that even suboptimal implementations using the pseudoinverse can still factorize a mechanism for $n = 10,000$ in 146 minutes on a V100 GPU, remaining well within practical requirements since we need only generate a mechanism once, before it can be reused indefinitely for training. In contrast, computing optimal matrices becomes practically difficult near $n \approx 10,000$, taking 24 hours to compute an effective factorization for $n = 8,192$ using batch-priority cloud CPU resources. We remark that this regime of $n$ is highly practical, e.g., standard federated benchmarks use $n \approx 2000$ (Reddi et al., 2020) and our central image classification use $n = 2000$. In terms of noise generation, the FFT mechanism shows preferable asymptotic properties, scaling as $\mathcal{O}(nd \log^2 n)$. However, even the optimal matrix mechanism with runtime scaling as $\mathcal{O}(dn^2)$, noise generation on a GPU (even with significantly suboptimal implementation) takes negligible time. Further, noise from our mechanisms can be pre-generated if needed. We discuss these tradeoffs in Appendix C.

## 5    EMPIRICAL EVALUATION

We compare four main mechanism classes: tree-based mechanisms (Honaker, 2015), including 'tree-completion' of Kairouz et al. (2021); our FFT mechanism; our optimal factorizations; and DP-SGD with (though incorrect) amplification via Poisson Subsampling (Abadi et al., 2016b).

When computing optimal factorizations by solving Eq. (6), we may vary the sensitivity constraint set $\mathcal{D}$ to encode different participation schemas. We compute optimal factorizations for the $(k, b)$-participation setting. An encoder $\mathbf{C}$ factorized for one value of $k$ can be applied for another, by simply computing its new sensitivity (due to our Section 2), though this will alter its privacy-utility tradeoffs. For example, our **MF1,6e** in Fig. 3 uses this to extend Denisov et al. (2022) to $k > 1$.

When applying a mechanism that is determined independently of the number of participations (e.g., FFT or tree aggregation) or extending an optimal mechanism for $k$ participations to a larger number of participations, the sensitivity may scale poorly in $k$. In such cases, it may actually have lower sensitivity to reuse a single encoder multiple times over the course of $n$ steps, and hence more favorable privacy-utility tradeoffs. We term this approach **encoder stamping**. This also provides a straightforward method for extending any factorization to handle more iterations without, e.g., re-optimizing Eq. (6). Combined with our Section 2, stamping lets us apply mechanisms from Denisov et al. (2022), e.g., **MF(k=1, n=1000)×2** in Fig. 1; mechanisms with stamping have "×$s$" appended in this way. Discussion of stamping and its relation to existing literature are in Appendix D.4.

The manner in which baselines from the extant literature map to this setting can be found in Appendix D.1. Since the matrix mechanism reduces privacy cost of training to that of the release of a single Gaussian mechanism, accounting in our case becomes quite simple; see Appendix D.2.

### 5.1    EXAMPLE-LEVEL DP FOR AN IMAGE CLASSIFICATION TASK.

We train image classification models on CIFAR10 (Krizhevsky, 2009) which has become a de facto standard for comparing DP ML algorithms—sufficiently easy for existing DP algorithms to achieve nontrivial accuracy, but not so simple so as to be unable to differentiate approaches. Details on our full setup are in Appendix E; generally, they match those of Kairouz et al. (2021). Notably, we make improvements on their Online Honaker-based approach by not just completing the tree with virtual steps, but also zeroing out noise from virtual steps as detailed in Appendix D.3. We find this led to significant improvements around a few percentage points. For all matrix mechanisms except the Denisov et al. (2022) baseline and our Optimal MF($k = 20, \dim = 2000) \times 1$, we optimize over the stamps $s$ by the losses in Appendix D.5, which we find well match the ordering in ML accuracy.

In contrast to Section 5.2, in this section we only compare factorizations of the prefix-sum matrix; we do not incorporate momentum or cooldown directly into the mechanisms, though we use both momentum and cooldown as postprocessing for the matrix-factorization-based mechanisms and report results for the best settings we find (with both). For DP-SGD, we report results for the best setting (no momentum, with cooldown). Details are in Appendix E.

**Main results (Figure 1)**    First, we see that the optimal factorization for the target $(k, b) = (20, 100)$ setting outperforms all other mechanisms across (nearly) all privacy levels, only slightly underperforming DP-SGD with amplification at $\varepsilon = 2$. To the best of our knowledge, this represents the first empirical demonstration of an ML algorithm which is competitive with DP-SGD into this high-privacy regime, *without any amplification by sampling*. The FFT (optimal decoder) mechanism outperforms all baselines, again well toward the high-privacy regime at $\varepsilon \approx 4$. Though at a worse privacy-utility tradeoff compared with our multi-epoch optimal matrices, this mechanism shows promise for outperforming prior work when $n$ grows too large for generating optimal factorizations.

### 5.2    USER-LEVEL DP FOR A NEXT WORD PREDICTION TASK

User-level DP for language models is an important real-word task (McMahan & Thakurta, 2022). There is much history in the DP language modelling literature, which we briefly describe in Appendix F.1. Here, we use the standard benchmark: StackOverflow next-word prediction (Reddi et al., 2020). Our experimental setup fixes the same model and hyperparameters as Kairouz et al. (2021) and Denisov et al. (2022) except notable changes below. Details are in Appendix F.2.

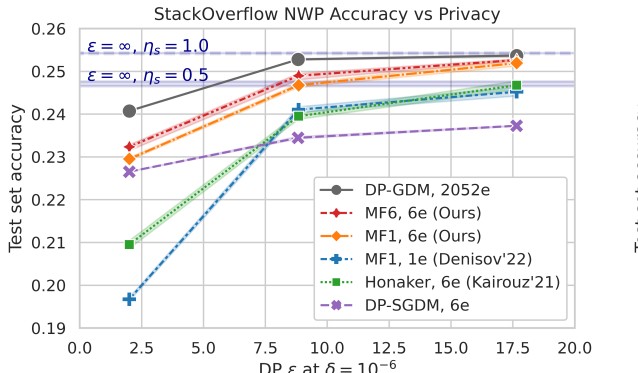 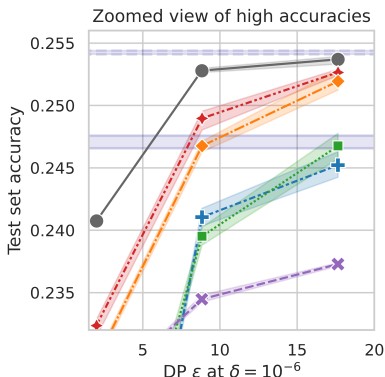

Figure 3: **Our MF6,6e achieves within** $2\%$ **relative difference in performance from the non-private baseline at** $\varepsilon = 8.8, \delta = 10^{-6}$. Our **MF1,6e**, though worse, outperforms all baselines from the literature. Select runs were replicated multiple times, with bootstrap $95\%$ intervals shown. **DP-GDM,2052e** is the extreme case of every-round participation ($342\times$ more computationally expensive than 6 epochs of training), and hence generally infeasible. The server learning rate $\eta_s$ was optimized over 0.5 and 1.0, and additionally 0.25 for $\varepsilon = 2$. The blue bands give the non-private baseline accuracy for 6 epochs of training with $\eta_s = 1.0$ and 0.5.

**Notable changes from prior work**  We use a much higher 1000 clients per round ($\approx 100$ in prior work)—enabled mainly by our multi-epoch factorizations. We also zero out large updates with $\ell_\infty$ norm greater than 100 (rather than scaling down to our clipping norm of $\zeta = 1$) as we found this improved the stability of noisy training (see Table 2 in Appendix F.3). This may have enabled more consistent success of a higher server learning rate $\eta_s = 1.0$.

We conducted initial simulations which verified that two observations from Denisov et al. (2022) for the single-epoch setting extended to our multi-epoch and large-batch (1000clients/round instead of 167) setting. First, linear server learning rate cooldown from $1\times$ to $0.05\times$ over the final 512 rounds offered a small improvement over constant rates (more so in the higher-privacy regime). Second, optimizing a factorizing with momentum and this cooldown schedule, rather than applying both as post-processing, consistently offers a small benefit. See Appendix F.2 for details. Thus, for our primary investigation we fix these preferable design choices and compare the following algorithms.

**Algorithms**  All algorithms train for 6 epochs 2052 rounds, and a large-batch 1000 clients/round (better for DP training) unless otherwise noted. **Honaker,6e** is the DP-FTRL algorithm of Kairouz et al. (2021), trained for 2048 rounds (a power of 2). **MF1,1e** (Denisov et al., 2022) is the state-of-the-art for single-epoch training, using 167 clients/round and $k$=1. **MF1,6e** uses our Eq. (3) to take the (non-negative) ($k$=1)-optimized matrix of the previous approach and compute the sensitivity under ($k$=6, $b$=342)-participation, allowing us to train for 6 epochs (2048 rounds) with large batches. **MF6,6e** is our approach directly optimizing the matrix factorization for ($k$=6, $b$=342)-participation via Eq. (6). **DP-SGDM,6e** is the `DP-FedAvg` algorithm of McMahan et al. (2018), to 2052 rounds, and accounted with Poisson sampling—an incorrect, though standard, accounting computation, as noted in Section 1. **DP-GDM,2052e** is full-batch gradient descent for 2052 rounds and 2052 epochs, or $342\times$ the computation cost of our 6 epoch runs. We compute the exact privacy cost for this approach, and estimate the accuracy from experiments with 1000 clients per round for computational efficiency, following the methodology of Kairouz et al. (2021). This is essentially an upper bound on the best privacy-accuracy tradeoffs with unlimited computational resources.

**Main results (Figure 3)**  We find that our **MF6,6e**, is the best feasible private result at $25.25\%$ accuracy and $(17.7, 10^{-6})$-DP. This exceeds the non-private baseline of $25.2\%$ accuracy reported by Kairouz et al. (2021), is within the margins of small hyperparameter tuning differences of our improved non-private baselines ($25.43\%$) and private full-batch gradient descent ($25.31\%$). At $(8.84, 10^{-6})$-DP, **MF6,6e** achieves $24.94\%$ accuracy, substantially improving over the previous state-of-the-art at this privacy level given by **MF1,1e** at $24.11\%$ accuracy, and considerably improves on both the accuracy and privacy of **Honaker,6e** ($24.86\%$ accuracy at $(21.0, 10^{-6})$-DP). In fact, we achieve better accuracy at $\varepsilon = 8.84$ than prior methods achieve at $\varepsilon = 17.7$. **DP-SGDM,6e** is outperformed by **MF6,6e** and **MF1,6e** across all $\varepsilon$ values evaluated. Fig. 12 in Appendix F shows results for each learning rate separately, with numeric results in Tables 4 and 5.

*Discussion and conclusions can be found in Appendix G.*

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

## A    SUMMARY OF NOTATION AND TERMINOLOGY

The following table summarizes the notation used throughout the paper:

| | |
|---|---|
| $n$ | Number of steps of the streaming linear query (SGD steps or FL rounds) |
| d | Dimension of per-step user contributions. |
| $\mathbf{x}_i \in \mathbb{R}$ or $\mathbb{R}^d$ | Sum of per-example gradients (or per-user model updates) on step $i$. |
| $\mathbf{x} \in \mathbb{R}^{n \times d}$ | Stream of inputs $\mathbf{x}_i$, equiv. matrix with rows $\mathbf{x}_i$ (so $\mathbf{x}_i = \mathbf{x}_{[i,:]}$). |
| $\zeta$ | Clipping norm that limits the size of per-example contributions to $\mathbf{x}_i$. |
| $\pi$ | Participation pattern, the set of steps that an example could participation in. |
| $\Pi$ | Participation schema, set of sets of steps (set of all $\pi$) an example could participate in. |
| $\mathfrak{D}$ | $= \{\mathbf{x} - \tilde{\mathbf{x}} \mid (\mathbf{x}, \tilde{\mathbf{x}}) \in \mathbf{N}\}$, the set of deltas between neighboring input streams $\mathbf{x}, \tilde{\mathbf{x}}$. |
| $\mathcal{D}$ | Corners of $\mathfrak{D}$ when assumed to be a polytope, $\mathfrak{D} = \mathrm{conv}(\mathcal{D})$. |
| $(k, b)$-participation | participation schema $\Pi$ with at most $k$ participations, separated by exactly $b$. |
| $\mathbf{A} \in \mathbb{R}^{n \times n}$ | Lower-triangular linear query matrix to be factorized as $\mathbf{A} = \mathbf{BC}$. |
| $\mathbf{T} \in \mathbb{R}^{n \times n}$ | $\mathbf{T} := \mathbf{A}^\top \mathbf{A}$ for convenience. |
| $\lambda_{\min}(\mathbf{A}), \lambda_{\max}(\mathbf{A}).$ | Smallest and largest eigenvalues of real matrix $\mathbf{A}$. |
| $\mathbf{A}^*$ | Conjugate (Hermitian) transpose of $\mathbf{A}$. |
| $\mathbf{X}^\star$ | A matrix $\mathbf{X}$ that is "optimal" in a context-dependent sense. |
| $\mathbf{A}^\dagger$ | Moore-Penrose pseudoinverse of matrix $\mathbf{A}$. |
| $\mathbf{A}_{[i,j]}$ | The $(i, j)^{\text{th}}$ entry of matrix $\mathbf{A}$. |
| $\mathbf{A}_{[i,:]}$ and $\mathbf{A}_{[:,j]}$ | The $i^{\text{th}}$ row and $j^{\text{th}}$ column. |
| $s$ | Number of encoder $\mathbf{C}$ replications (stamps) into a block-diagonal matrix. |
| $\mathrm{conv}(S)$ | Convex hull of the set $S$. |
| $[n]$ | $= \{1, \ldots, n\}$ |
| $\|\mathbf{X}\|_F$ | The Frobenius norm of a matrix $\mathbf{X}$. |

We utilize terminology from federated learning as well as standard centralized training, which generally map as follows:

| Centralized | Federated |
|---|---|
| example | user or client |
| batch size | clients-per-round |
| DP-SGD | DP-FedAvg |
| step or iteration | (communication) round |
| gradient | model update |

## B    GENERALIZED SENSITIVITY AS AN OPERATOR NORM

Eq. (2) shows that our generalized notion of sensitivity can be viewed directly as a particular operator norm. To see this, view $\mathbf{C} : \mathbf{V}_1 \to \mathbf{V}_2$ as a linear operator from vector space $\mathbf{V}_1$ to $\mathbf{V}_2$. Then with $\| \cdot \|_{(1)}$ the vector norm on $\mathbf{V}_1$ and similarly for $\mathbf{V}_2$, an operator norm is defined as

$$\|\mathbf{C}\|_{(1),(2)} = \max_{u \in \mathbf{V}_1 : \|u\|_{(1)} \leq 1} \|\mathbf{C}u\|_{(2)}.$$

Because we use the Gaussian mechanism and thus are interested in the $\ell_2$ sensitivity, $\|\cdot\|_{(2)} = \|\cdot\|_2$, and we define the norm

$$\|\mathbf{u}\|_{(1)} = \|\mathbf{u}\|_{\mathfrak{D}} := \inf\left\{ r > 0 : \frac{\mathbf{u}}{r} \in \mathfrak{D} \right\},$$

the vector norm induced by $\mathfrak{D}$ (the fact that $\mathfrak{D}$ is a closed, convex, symmetric set ensures this is a norm). Note $\mathbf{u} \in \mathfrak{D} \Leftrightarrow \|\mathbf{u}\|_{\mathfrak{D}} \leq 1$. Thus, we have

$$\mathrm{sens}_{\mathfrak{D}}(\mathbf{C}) = \|\mathbf{C}\|_{\mathfrak{D},2}. \tag{9}$$

## C   THE FFT MECHANISMS AND REDUCING COMPUTATION

---

**Algorithm 1** DP-Prefix Sum Computation via FFT (with $d = 1$)

---

**Inputs:** Data vector $\mathbf{x} \in \mathbb{R}^n$ (with each $|\mathbf{x}_i| \leq \zeta$) and zCDP parameter $\rho$.

$\mathbf{v}^{\mathsf{DFT}} \in \mathbb{C}^{2n} \leftarrow$ the DFT of $\mathbf{v}$ (defined in Eq. (34)). Let $\mathbf{v}^{\mathsf{DFT}}_{[:n]}$ be the first $n$ coordinates.

$\mathbf{F} \leftarrow$ DFT matrix in $2n$-dimensions, where the $k$-th row of $\mathbf{F}$ is given by

$$\mathbf{F}_{[k,:]} = \frac{1}{\sqrt{2n}}\left[\exp\left(-\frac{j2\pi ka}{2n}\right) : a \in \{0, \ldots, 2n-1\}\right].$$

$(\Sigma, \mathbf{w}) \leftarrow (\mathrm{diag}(\mathbf{v}^{\mathsf{DFT}}),$ standard complex Normal in $2n$-dimensions$)$.

$(\mathbf{s}, \widetilde{\mathbf{z}}) \leftarrow \left(\left[\mathbf{x}_0, \mathbf{x}_0 + \mathbf{x}_1, \ldots, \sum_{a=0}^{n-1} x_a\right], \sqrt{\frac{\kappa^2 \|\mathbf{v}^{\mathsf{DFT}}\|_1}{4n\rho}}\left(\mathbf{F}^* \Sigma^{1/2} \cdot \mathbf{w}\right)\right).$

**Output** $\mathbf{s} + \widetilde{\mathbf{z}}.\mathtt{real}[0, \ldots, n-1].$

---

We propose two FFT mechanisms. First, we propose the FFT mechanism which is described in Algorithm 1. This mechanism has the same computation complexity—no optimization costs and $\mathcal{O}(nd \log n)$ noise generation—as the Honaker method used in Kairouz et al. (2021) but at a better privacy-utility tradeoff, as shown in Fig. 9. The privacy and utility analysis can be found in Appendix J.

The FFT Optimal Decoder (FFT_Opt_Dec) mechanism presented in Section 4 represents taking (a real-valued translation of) the encoder $\mathbf{C}$ from Algorithm 1 and using the optimal decoder, defined in terms of the Moore-Penrose pseudoinverse of $\mathbf{C}$. Similarly to Algorithm 1, there is no need to construct a literal matrix to multiply by in the case of noise defined by the optimal decoder; noise generation time of the mechanism, however, increases by a logarithmic factor to $\mathcal{O}(nd \log^2 n)$ (as discussed in Appendix K). This complexity is still feasible for many steps (which we will discuss below) and comes with significant utility benefits (see Fig. 1).

All the mechanisms we study scale as either $\mathcal{O}(n^2)$ or $\mathcal{O}(n \cdot \mathrm{polylog}(n))$. For our $n = 2000$ step environments, and even far beyond to $n \approx 10,000$, our algorithms can be efficiently realized on GPUs with runtime on the order of seconds per step (including computing and applying gradients and noise). The main challenge in these cases are storing the $n^2 + nd$ coordinates in GPU memory (the former for the decoder matrix, the latter for the noise samples). Given that each of the $d$ coordinates of noise can be sampled independently, this algorithm is straightforwardly parallelizable and so work may be partitioned across many processors when needed. Noises could instead be pre-generated in an entirely separate process, and stored on disk, to be loaded into memory row-by-row concurrently with training.

Outside of computation, both our FFT mechanisms take $O(nd)$ space as all noises for the $\mathbf{x} \in \mathbb{R}^{n \times d}$ must be pre-generated. This is in contrast to all prior work, and even our optimal factorizations, which require only $\mathcal{O}(d)$ space to generate the noise at the current step. Again, we note that space is ofter much cheaper so this is typically not the limiting factor.

# D MECHANISMS UNDER CONSIDERATION: BASELINES, SUBTLETIES, AND LOSSES.

## D.1 BASELINES

Kairouz et al. (2021) and Denisov et al. (2022) both present approaches for ML model training which can be understood as instances of the matrix mechanism–the former grounded in the binary-tree mechanism as refined by Honaker (2015), and the latter explicitly optimizing a factorization under single participation. These two works yield two natural baselines:

- Kairouz et al. (2021) explores 'tree restarts' (generalized as our notion of 'stamps', $s$, in Section 5) and the so-called 'tree-completion trick' for the Honaker estimator-from-below variant of the binary tree method for computing differentially private prefix sums; the matrix-factorization perspective on this estimator allows us to implement slightly optimized versions of these methods; see Appendices D.3 and D.4.

- Denisov et al. (2022) computes optimal factorizations of various optimization-related matrices, though only for a single epoch. For these matrices, we leverage the results in Section 2 to directly compute the sensitivity of the encoder matrices for multiple participations.

These two papers can be combined in other ways as well; e.g., the results of Denisov et al. (2022) show that the 'fully efficient estimator' of Honaker (2015) may be used as a drop-in replacement for the estimator from below in Kairouz et al. (2021). We focus on the two mechanisms specified above as the natural baselines for the present work.

## D.2 PRIVACY ACCOUNTING

The matrix mechanism Eq. (1) conceptually adds isotropic Gaussian noise in some encoded space. In our case, we encode a matrix of gradients (clipped to $\ell_2$ norm $\zeta$) computed over the course of training, denoted by $\mathbf{G}$, with the matrix $\mathbf{C}$, and add Gaussian noise to each entry in the matrix $\mathbf{CG}$. Under the assumption that the matrix factorization has been appropriately scaled so $\mathbf{C}$ has sensitivity 1, this Gaussian noise will have standard deviation $\sigma = \zeta z$ in each coordinate, where $z$ is the 'noise multiplier' parameter determining the privacy level of the mechanism (see Table 3 for example).

Privacy costs are computed as a single application of the Gaussian mechanism to $\mathbf{GC}$ using the `PLDAccountant` provided by the Google DP Library[3]. We also use this accountant to analyze DP-(S)GD baselines (which require more complex accounting), yielding a small improvements in $\varepsilon$ over the Renyi-DP accounting used in prior works.

## D.3 IMPROVEMENTS TO 'TREE COMPLETION' BY REMOVING NOISE FROM VIRTUAL STEPS

The "tree completion" trick of Kairouz et al. (2021) is used on the last step of any restart (in ours, stamp) to reduce the noise added on this step. This is achieved by adding virtual steps (with 0 inputs) until the final step of that level in the tree, because this noise will be the lowest in that level. In this section, we show how to further improve on this trick and that our matrix mechanisms make analyzing such tricks easier. Our implementations of the binary-tree baslines Honaker (2015); Kairouz et al. (2021) utilize these improvements.

For the online Honaker estimate, Honaker (2015) obtain a DP estimate for the release node $i \in [n]$ (representing the prefix sum until $i$) but summing the corresponding subtrees prior to this node. These are exactly the subtrees corresponding to the binary representation of this node (Honaker, 2015). Then, the variance required to release node $i$, with subtrees of height $0, 1, ..., l_i - 1$, is

$$\left(\sum_{j=0}^{l_i-1} c_j^2 \cdot 2^j\right) \cdot \sigma^2 = \frac{1}{2 \cdot (1 - 2^{-\mu})} \cdot \sigma^2 \qquad \text{where} \qquad c_j = \frac{1/2^j}{\sum_{j=0}^{l_1-1} (1/2^j)}.$$

Notice that reaching a new height in the tree decreases the variance needed. In Kairouz et al. (2021) and just before terminating on some non-power-of-two step $n' < n$, they run $n - n'$ virtual steps

---

[3]https://github.com/google/differential-privacy

on zero gradients. This enables their mechanism to use the minimal noise for the power-of-two-step for the final real step.

However, notice that in both these cases, the methods assume that these virtual steps must be privatized. Indeed, they do not need to be because we know apriori that these steps are virtual, i.e., not corresponding to real gradients. Thus, in our methods we account for this in our mechanism and reduce the noise of the final step accordingly. Importantly, this can be computed without altering the asymptotic runtime and storage complexity of the algorithm: on the last step, the contributions of the virtual steps to the power-of-two noise can be calculated using, e.g., a second binary tree, and removed. This leads to a significant benefit in the loss as we observed in Table 1 in Appendix D.5.

We believe this oversight of prior works showcases the power of our matrix mechanism approach. Indeed, let $\mathbf{C}_{\text{tree}}$ be a matrix representing the (complete) binary tree used in the mechanisms of Honaker (2015); Kairouz et al. (2021) with $2^{\lceil \log_2(n) \rceil}$ leaves; for the sake of concreteness, assume this is the matrix constructed in Appendix C of Denisov et al. (2022). Let $\mathbf{B}_{\text{Hon}}$ represent the Honaker estimator-from-below; in our language, the decoder used by (Kairouz et al., 2021).

The tree completion trick of (Kairouz et al., 2016) can be understood as follows. The matrix $\mathbf{B}_{\text{Hon}}\mathbf{C}_{\text{tree}}$ is of size $2^{\lceil \log_2(n) \rceil} \times 2^{\lceil \log_2(n) \rceil}$. In the case that $n$ is not a power of two, the penultimate rows and columns of this matrix will go unused. However, for this factorization, the variance added by $\mathbf{B}_{\text{Hon}}$ on the final row will be quite small, due to the binary tree's redundancy in encoding estimates of this sum. The matrix we wish to factorize is a prefix-sum matrix $\mathbf{S}$ of size $n \times n$; this matrix can be expressed as any one of a family of transformations of the (potentially larger) product $\mathbf{B}_{\text{Hon}}\mathbf{C}_{\text{tree}}$:

$$\mathbf{S} = \mathbf{P}_j \mathbf{B}_{\text{Hon}} \mathbf{C}_{\text{tree}} \mathbf{E},$$

where $\mathbf{E}$ embeds a $n$-dimensional vector into $2^{\lceil \log_2(n) \rceil}$ dimensions by padding with zeros, and $\mathbf{P}_j$ projects back down to $n$ dimensions in a similarly axis-aligned way, taking the first $n-1$ rows of its right-hand matrix argument, and only one, but *any* of the $j^{th}$ rows for $n \leq j \leq 2^{\lceil \log_2(n) \rceil}$. To minimize the Frobenius norm of the constructed decoder in the factorization of $\mathbf{S}$, we may simply pick the row with the lowest $\ell_2$ norm; in the case of $\mathbf{B}_{\text{Hon}}$, this is the final row.

One more optimization becomes clear when tree completion is formulated in this manner. Similar to our optimal decoder of Section 4, any decoder can be used without changing the sensitivity of the encoder. Noting that nonzero entries in the decoder increase our loss of Eq. (4), we can simply zero out the columns of the decoder corresponding to these virtual steps—this decreases the loss, preserves the same error in the DP estimate of the prefix sum (the inputs are 0), and maintains the same DP guarantee. In other words, we need not account for the noise, or the error it introduces, of virtual steps. We now provide a more rigorous explanation.

The image of $\mathbf{C}_{\text{tree}}\mathbf{E}$ can be contained in an axis-aligned subspace; effectively, the subspace corresponding to elements that may be nonzero in the binary tree when run for $n$ steps. In other words, the columns of the decoder corresponding to rows of the encoder that are removed via the projection need not be included: because the input is not processed. Therefore, denoting the projection onto this subspace by $\Pi$, we may write:

$$\mathbf{S} = \mathbf{P}_j \mathbf{B}_{\text{Hon}} \Pi \mathbf{C}_{\text{tree}} \mathbf{E} = (\mathbf{P}_j \mathbf{B}_{\text{Hon}} \Pi)(\Pi \mathbf{C}_{\text{tree}} \mathbf{E}),$$

further reducing the variance of the decoder (without increasing the sensitivity of the encoder) in this incomplete binary tree case.

In our implementations of the online Honaker mechanism, we freely use these tricks, in addition to exact calculations of the sensitivity of the mechanism enabled by noting that the encoder is all-nonnegative and the observations of Section 2, leading to some improvements in these mechanisms over those in the existing literature. No changes to accounting are required, as privacy is inherited from the matrix mechanism perspective.

### D.4 STAMPING: REPEATED MECHANISMS IN THE MATRIX-FACTORIZATION SETTING

In stamping, we define a new encoder matrix as the Kronecker product of some given encoder $\mathbf{C}$ with an $s \times s$ identity matrix $\mathbf{I}$, creating a new $sn$-dimensional linear DP query mechanism. Assuming $\mathbf{C}$ is of shape $n \times n$, the resulting encoder is an $sn \times sn$ block-diagonal encoder matrix, formed by 'repeating' the matrix $\mathbf{C}$ along the diagonal.

Kairouz et al. (2021) explored 'restarting' their binary-tree based prefix sum estimation mechanism, treating the number of restarts used as a hyperparameter, and treating the result of a 'completed' application of the binary tree as fixed. For linear operators $\mathbf{A}$ with constant columns below the diagonal, this approach may be used to construct a new factorization from an existing one. This constant-columns property, or a block-based variant thereof, is *required* to simply treat the output from a 'completed' application of the existing mechanism as fixed; for a general matrix $\mathbf{A}$, there is no clear prefix property which can be leveraged for this purpose.

Taking the matrix view, one may construct a similar encoder/decoder pair for the prefix-sum matrix by reusing the initial decoder $\mathbf{B}$ on the block-diagonal and fixing the columns to simply repeat the final row of this decoder below the block-diagonal; notice that the constant-column property of the prefix sum matrix guarantees that this construction appropriately factorizes $\mathbf{A}$. The noise that the matrix mechanism thus constructed adds can be implemented as a 'restarted' tree mechanism; however, since we compute sensitivities exactly for decoders of this structure as in Section 2, the privacy properties of these mechanisms we construct to replicate the 'restarts' of (Kairouz et al., 2021) may not be identical to those presented there, where accounting is performed by composition.

The matrix-mechanism perspective additionally allows one to apply the 'stamping' construction to *any* linear operator $\mathbf{A}$ (e.g. the momentum matrix), where a reuse of fixed previous outputs is not possible. A 'stamped' factorization of *any* $\mathbf{A}$ may be obtained, for example, by matrix pseudoinversion: letting $\mathbf{B} = \mathbf{A} \left( \mathbf{C} \otimes \mathbf{I} \right)^{\dagger}$. This defines a legitimate factorization of any $\mathbf{A}$ requiring only suitable non-degeneracy assumptions on $\mathbf{C}$, and indeed represents the optimal decoder for the stamped encoder.

The pseudoinverse-based construction can, even in the prefix-sum case, be *quite different* from a construction designed to replicate 'restarted' mechanisms. For example, if $\mathbf{C}$ is a matrix representation of the binary-tree encoder and $\mathbf{I} = 1$, then the resulting decoder matrix represents the *full*, rather than online, Honaker decoder (Honaker, 2015); the validity of this mechanism in the adaptive streaming setting was only shown quite recently by Denisov et al. (2022).

For comparability with existing literature (and to preserve potential for an efficient implementation), however, all of the tree-based mechanisms we explore in the main body have decoders which replicate the setting of composition[4]. All other stamped mechanisms used the optimal decoder.

**Optimizing over** $s$  Considering instantiation enables us to directly analyze and minimize (over $s$) the stamped mechanism's multi-epoch loss Eq. (4) without running compute intensive ML experiments. Indeed, we observe in Table 1 of Appendix D.5 that for mechanisms which were not explicitly optimized for the $(k, b)$ participation setting, there exist stamped mechanisms ($s > 1$) with much lower loss that correspondingly led to much more performant ML models (e.g., Figures 7 and 8 of Appendix D).

Interestingly, with our capacity to measure mechanisms at a single shot in the multi-epoch setting, we see a similar trend as was observed for restarts in Kairouz et al. (2021): that 'stamped' mechanisms have lower total loss than their non-stamped full-tree counterparts in the 20-epoch, 100 steps / epoch setting (see Table 1); training performance of these 'stamped' mechanisms on CIFAR10 can be found in Fig. 7. However, there is a significant improvement in our approach in that we can now directly tune this hyperparameter without the need to actually run ML training. This lets us reduce computation by only analyzing the loss of the generated matrices and then running the mechanism with the lowest loss.

---

[4] We show how the optimal binary-tree decoder differs from the online, composition-based decoder in Fig. 6

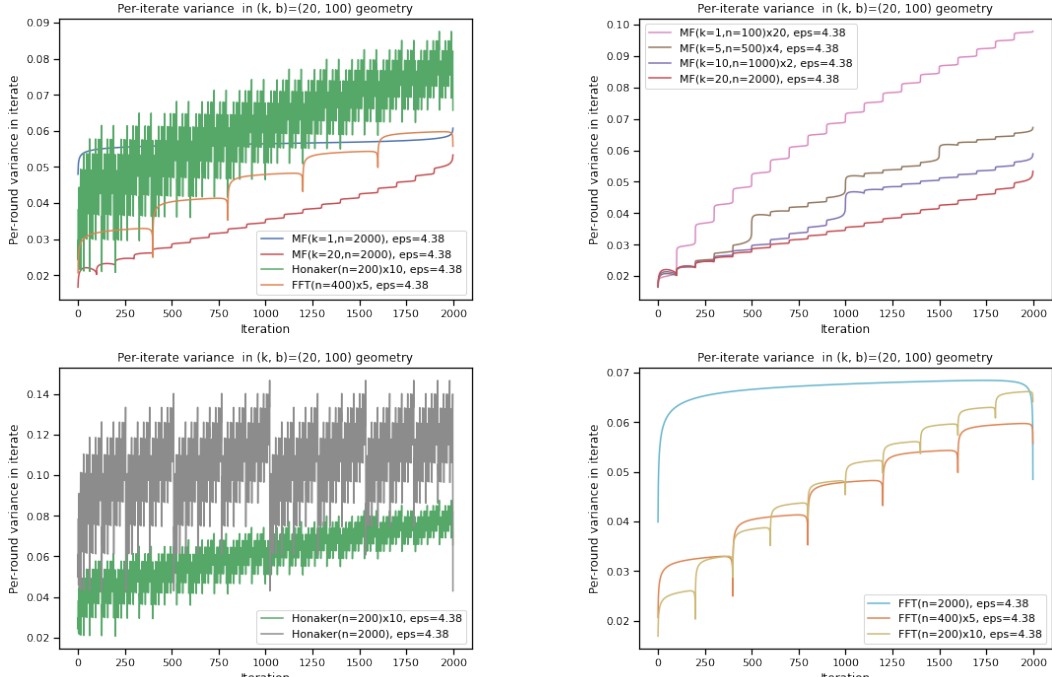

Figure 4: Per-iterate variance for prefix-sum factorizations. All mechanisms above yield the same privacy ($(\varepsilon, \delta) = (4.38, 10^{-5})$ in the $(k, b) = (20, 100)$ setting), but have different total variances (the integral of the curves above).

### D.5 Factorization losses and per-iterate variance

As a first measure of the privacy-utility tradeoff, we compare the losses of each mechanism from Eq. (4) for factorizations of the matrices under consideration.

We compare measured losses of several factorizations of the prefix-sum matrix for the $(k, b) = (20, 100)$ setting of the CIFAR experiments in Table 1. We plot the per-iterate variance of the mechanisms in Fig. 1, along with several variations, in Fig. 4. In Fig. 5, we plot the per-iterate variance distribution of factorizations of the momentum and cooldown matrix (described in Section 5) at a fixed variance level, and with various privacies, computed for the $(k, b) = (6, 342)$ participation setting.

Figs. 4 and 5 both demonstrate the effect of $(k, b)$-participations on the optimization problem. Particularly interesting to consider are the optimally-factorized matrices; in both cases, the epoch structure is clearly visible in the manner in which the mechanisms distribute variance. We see also the effect of 'stamps' $s$ in the variance distribution, effectively a proxy for the epoch structure directly accounted for by the optimal mechanisms.

| Mechanism | $(k, b) = (20, 100)$ Prefix Sum Loss | Computation for $n$ Steps, $d = 1$ |
|---|---|---|
| (Online) Honaker($n = 2000$) | 5.8e6 | $\mathcal{O}(n \log n)$ Noise Generation |
| (Online) Honaker($n = 1000$)$\times 2$ | 3.3e6 | |
| (Online) Honaker($n = 400$)$\times 5$ | 2.1e6 | |
| **(Online) Honaker(n = 200)$\times$10** | **2.0e6** | |
| (Online) Honaker($n = 100$)$\times 20$ | 2.1e6 | |
| Optimal Decoder Honaker($n = 2000$) | 2.4e6 | $\mathcal{O}(n^2)$ Noise Generation |
| Optimal Decoder Honaker($n = 1000$)$\times 2$ | 1.6e6 | |
| **Optimal Decoder Honaker(n = 400)$\times$5** | **1.2e6** | |
| Optimal Decoder Honaker($n = 200$)$\times 10$ | 1.4e6 | |
| Optimal Decoder Honaker($n = 100$)$\times 20$ | 1.8e6 | |
| **MF(k = 20, n = 2000)** | **6.5e5** | $\mathcal{O}(n^3)$ Optimization + $\mathcal{O}(n^2)$ Noise Generation |
| MF($k = 10, n = 1000$)$\times 2$ | 8.8e5 | |
| MF($k = 5, n = 500$)$\times 4$ | 1.2e6 | |
| MF($k = 1, n = 100$)$\times 20$ | 2.5e6 | |
| MF($k = 1, n = 2000$) | 1.6e6 | |
| **MF(k = 1, n = 1000)$\times$2** | **1.37e6** | |
| MF($k = 1, n = 500$)$\times 4$ | 1.4e6 | |
| MF($k = 1, n = 400$)$\times 5$ | 1.5e6 | |
| MF($k = 1, n = 200$)$\times 10$ | 1.8e6 | |
| MF($k = 1, n = 100$)$\times 20$ | 2.5e6 | |
| FFT($n = 2000$) | 2.3e6 | $\mathcal{O}(n \log n)$ Noise Generation |
| FFT($n = 1000$)$\times 2$ | 1.8e6 | |
| **FFT(n = 400)$\times$5** | **1.6e6** | |
| FFT($n = 200$)$\times 10$ | 1.9e6 | |
| FFT($n = 100$)$\times 20$ | 2.5e6 | |
| FFT Optimal Decoder($n = 2000$) | 2.2e6 | $\mathcal{O}(n \log^2 n)$ Noise Generation |
| FFT Optimal Decoder($n = 1000$)$\times 2$ | 1.5e6 | |
| **FFT Optimal Decoder(n = 400)$\times$5** | **1.1e6** | |
| FFT Optimal Decoder($n = 200$)$\times 10$ | 1.2e6 | |
| FFT Optimal Decoder($n = 100$)$\times 20$ | 1.7e6 | |

Table 1: Loss for various prefix-sum factorizations, computed via Eq. (4), in multiple-participation setting for 20 epochs with 100 steps per epoch. Lowest-loss mechanism in each class bolded. Note that '(Online) Honaker' corresponds to the restarted decoder. By evaluating the dual problem (Section 3), 6.53e5 represents a lower bound on the optimal loss; the optimal matrix factorization is within 0.2% of this optimal value. Though up to $\mathcal{O}n^2$ noise generation can be tolerated practically for large ML training runs, we find that the stamped FFT optimal decoder obtains the best privacy-utility tradeoffs while requiring only $\mathcal{O}(n \log^2 (n))$ time. Sensitivity is calculated exhaustively with contributions constrained to $+1$ for all matrices except FFT ones, where sensitivity is calculated using Theorem 2.1.

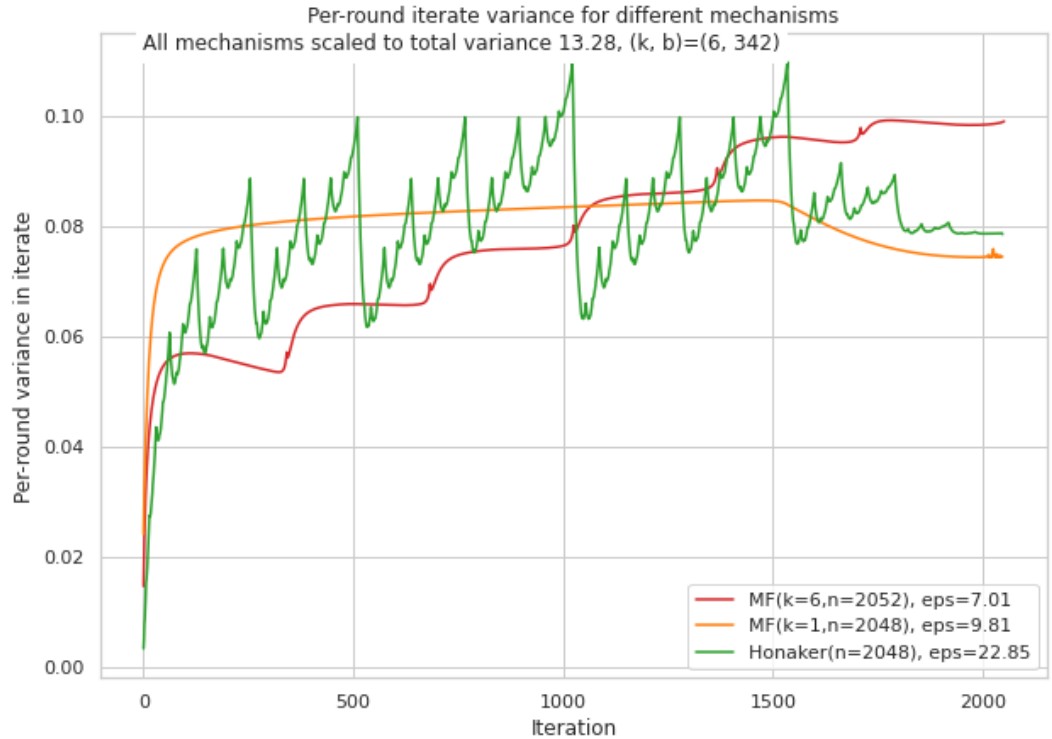

Figure 5: Per-iterate variance for momentum + cooldown matrix factorizations. Privacy measured in the $(k, b) = (6, 342)$ setting.

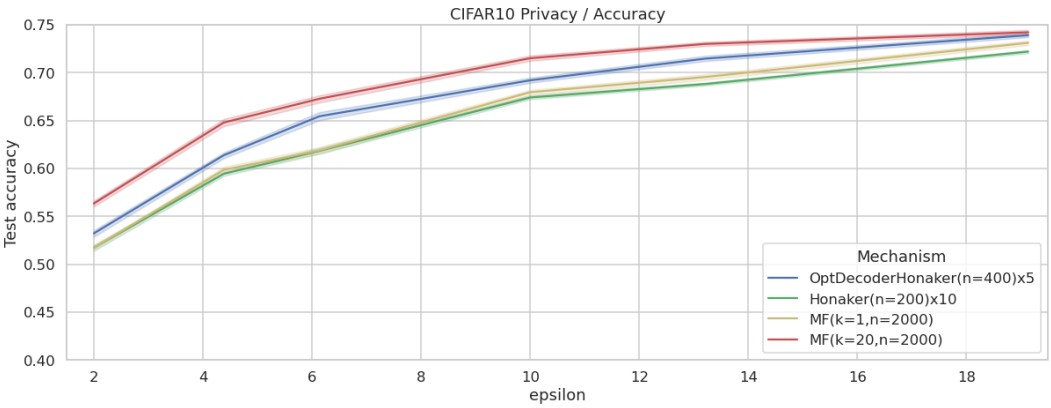

Figure 6: Comparing the optimal decoder, i.e., **OptDecoderHonaker**, with the standard stamping decoder (including fixing the output of each block), i.e., **Online Honaker**, with the optimal factorization.

# E DETAILS AND ADDITIONAL EXPERIMENTS FOR CIFAR10.

We train image-classification models using the CIFAR10 dataset as hosted in `tensorflow-datasets`, containing 50,000 training and 10,000 test examples. We evaluate and compute test accuracies on the entire test set, following the open-sourced code of Kairouz et al. (2021). We reuse the network architecture, dataset processing and initialization strategies presented in Kairouz et al. (2021); in particular, the architecture we use can be found in their Table 2 (b).

**Optimization setup and hyperparameters** We train all mechanisms for 20 epochs with batch size of 500, yielding 100 steps per epoch and 2000 total. After performing some small initial grid searches, we settled on using linear learning rate cooldown to $0.05\times$ the initial learning rate over the last 500 steps of training. We found this consistently improved utility for all mechanisms and privacy levels.

As mentioned in Section 5, for this 20-epoch training setup, we only compare factorizations of the prefix-sum matrix, and do not include any factorizations of matrices which incorporate momentum of learning rate cooldown directly in the mechanism itself (Denisov et al., 2022). We sweep over learning rates of values $(1 \times 10^i, 2 \times 10^i, 5 \times 10^i)$ for $i$ in $\{-2, -1\}$; for all mechanisms and noise levels, optimal values were in the interior of this sweep. We sweep over momentum values of $0, 0.85, 0.9, 0.95$ though find nonzero momentum works best for all matrix mechanisms, and no momentum works best for DP-SGD at our scale as found previously by Kairouz et al. (2021).

For Honaker and FFT-based factorizations, there is no known a-priori way to choose the optimal number of $s$ for a given $(k, b)$ setting. Therefore we treat the value $s$ as a hyperparameter, and sweep across it, for $s \in \{1, 2, 5, 10, 20\}$. As can be seen in Table 1, the optimal $s$ for both of these factorizations was in the interior of this sweep. As shown, e.g., in Fig. 7, the training-time performance of these mechanisms matched the expected order for computed loss. This value $s$ represents an extra hyperarameter which must be set for the Honaker and FFT mechanisms; to the best of our knowledge, computing the loss for various instantiations of these mechanisms via Eq. (4) represents the only known method for setting this parameter other than simply training models.

We also apply our sensitivity analysis of Section 2 to the matrices of Denisov et al. (2022) which are optimized for $k = 1$. In doing so, we can also optimize the number of stamps which we do. We report the best results as identified by the losses in Table 1.

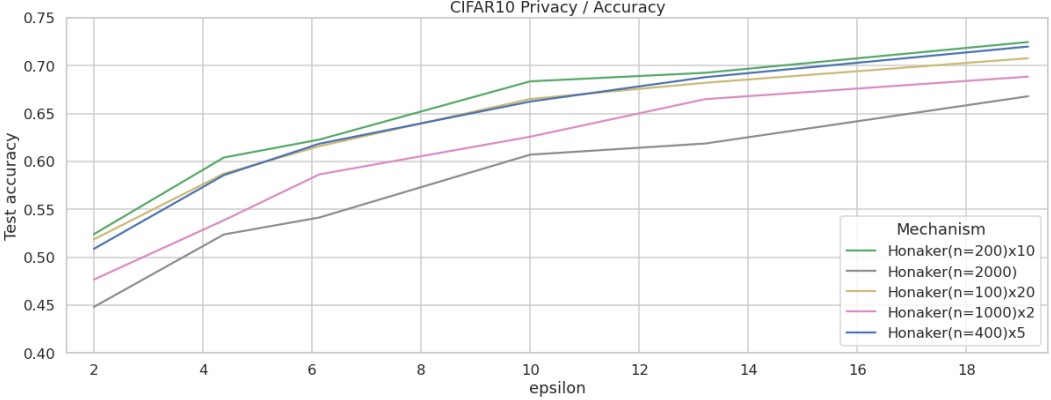

Figure 7: DP-FTRL-Honaker baseline ablation with respect to number of 'stamps' $s$.

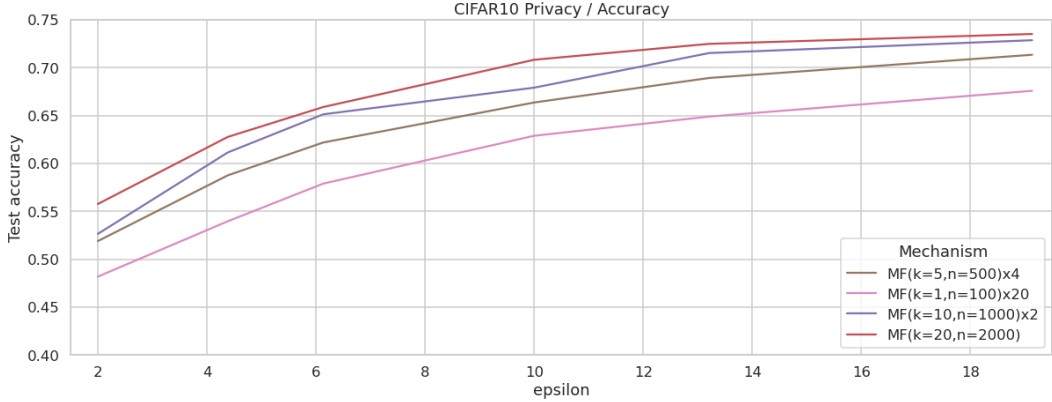

Figure 8: Ablation of prefix-sum factorizations, optimized for different number of epochs, and 'stamped' as appropriate. Performance improves as the geometry used for computing the factorization approaches that used for training.

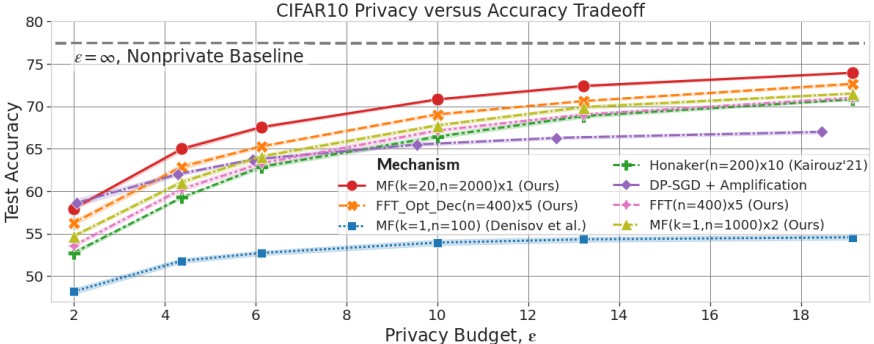

Figure 9: **Our optimal multi-epoch matrix and FFT-based mechanisms outperform all others, including DP-SGD with amplification**, as low as $\varepsilon \approx 4$. Using our sensitivity calculation of Theorem 2.1 and stamping (Section 5), we optimize a single pass ($k = 1$) matrix of Denisov et al. (2022) but apply it here with $> 1$ pass. We use an online Honaker-based decoder equivalent to that of Kairouz et al. (2021) except for a significant improvement to tree-completion in Appendix D.3. Models trained for 20 epochs on CIFAR10 with a batch size of 500. We repeat each setting 12 times and show 95% bootstrapped confidence intervals. Empirical setup is in Section 5.1.

# F    ADDITIONAL STACKOVERFLOW DETAILS

## F.1    PRIVACY AND LANGUAGE MODELLING

Language models trained on user data are an important real-world application of DP training, as these models can memorize their training data if appropriate mitigations are not applied (Carlini et al., 2019; Song & Shmatikov, 2019; Carlini et al., 2021). Since one user might contribute 1000s of tokens (training examples) to a dataset, it is particularly important to consider user-level guarantees (McMahan et al., 2018). Building on the approach of Kairouz et al. (2021), Google recently announced the first-ever launch of a language model trained on user data with a formal user-level DP guarantee ($\rho = 0.81$ zCDP), further demonstrating the importance of this application (McMahan & Thakurta, 2022).

The StackOverflow next-word prediction task, introduced in (Reddi et al., 2020), has become a benchmark problem for DP training, and our experimental setup here fixes the same model and adapts hyperparmaeters from previous work including Kairouz et al. (2021); Denisov et al. (2022).

## F.2    HYPERPARAMETER TUNING AND INITIAL EXPERIMENTS

All runs use server momentum 0.95 and a learning-rate cooldown schedule for the last 25% of rounds. Zeroing outlier updates and using 1000 clients/round (6 epoch runs) allows the use of the higher server learning rates. **MF1,1e** replicates the result of single-epoch training from Denisov et al. (2022); note that with 167 clients/round and this mechanism, the higher learning rate does not appear to help. Fig. 10 gives preliminary experimental results which informed the main experiments used in the paper. Note that the y-axis range (Test set accuracy) is highly compressed, and so the primary point of comparison is on epsilons. For example, Denisov et al. (2022) shows that cross-run variation of 0.002 or more is typical.

The 6 horizontal lines give test-set accuracy for various non-private training mechanisms. The "Un-noised MF" runs correspond to the same code path used for privacy, but without any noise addition. In particular, these use momentum with learning rate cooldown; the other unnoised runs use a standard FL implementation with momentum but a fixed learning rate schedule; "cpr=167" corresponds to one epoch of training (167 clients/round), and "cpr=50" is 50 clients/rounds (only about 1/3 of an epoch). This last non-private baseline uses the best hyperparameters for FedAvgM from Reddi et al. (2020).

The two "Unnoised MF" runs with accuracies between 0.246 and 0.248 are functionally identical, and the line near 0.248 accuracy is the same except it does not use learning-rate cooldown. Thus, for the given learning rates, we see the higher-epsilon private runs are adding sufficiently small noise that the accuracy is essentially equivalent to unnoised baselines with the same hyperparameters. However, using larger learning rates can achieve accuracy over 25%, even with privacy as in the case of the MF-6-6 run, hence motivating the inclusion of larger learning rates in the main experiments.

The MF (Matrix Factorization) runs with "prefix" in the name correspond to computing an optimal factorization of the prefix-sum matrix (lower triangular matrix of ones) and then applying momentum (and possibly learning rate cooldown) as post-processing. The other MF runs directly factor the momentum or momentum+cooldown matrix.

## F.3    IMPACT OF ZEROING-OUT LARGE-NORM UPDATES

We observed that zeroing out updates with an $\ell_\infty$ norm greater than $100\zeta = 100$ greatly stabilized training, allowing larger learning rates, particularly for **MF6,6e**. We conducted ablation experiments where we turned off this zeroing, which produced a large fraction of unconverged runs as detailed in Table 2. The number of updates zeroed increases significantly with larger amounts of noise and larger learning rates, as shown in Fig. 11.

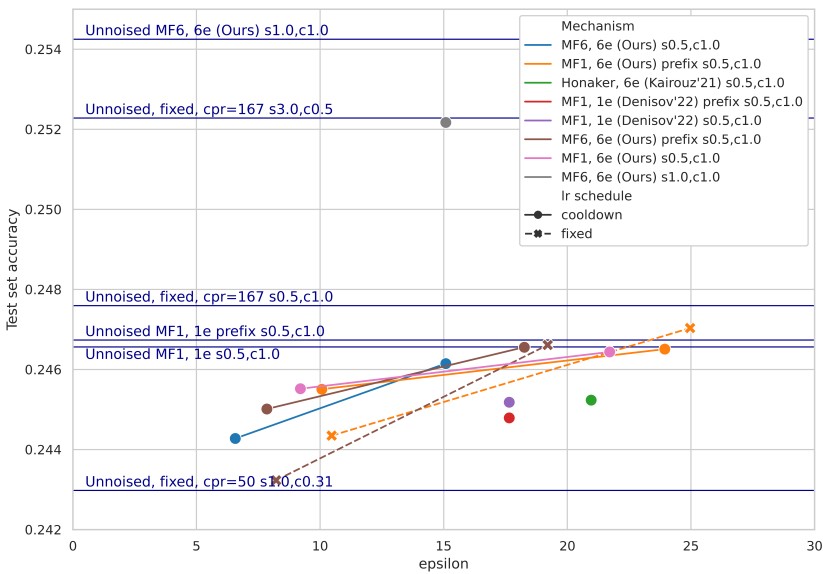

Figure 10: Preliminary experimental results and non-private (unnoised) baselines. The notation $sX, cY$ indicates a server learning rate $X$ and client learning rate $Y$.

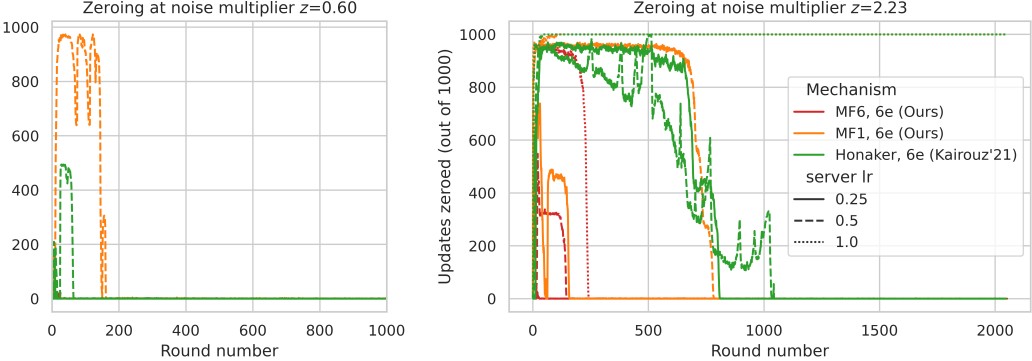

Figure 11: Number of large-magnitude updates zeroed per training round.

|  |  | Unconverged runs with | |
| Mechanism | $\varepsilon$ | Zeroing | No Zeroing |
|---|---|---|---|
| **DP-SGDM,6e**, $\eta_s = 0.5$ | 2 | 0 of 3 | 1 of 2 |
| **DP-SGDM,6e**, $\eta_s = 0.5$ | 9 | 0 of 3 | 0 of 2 |
| **MF6,6e**, $\eta_s = 0.5$ | 2 | 0 of 3 | 3 of 4 |
| **MF6,6e**, $\eta_s = 1.0$ | 9 | 0 of 3 | 3 of 4 |

Table 2: Number of divergent training runs with and without zeroing of user updates with $\ell_\infty$ norm greater than 100; $\eta_s$ gives the server learning rate.

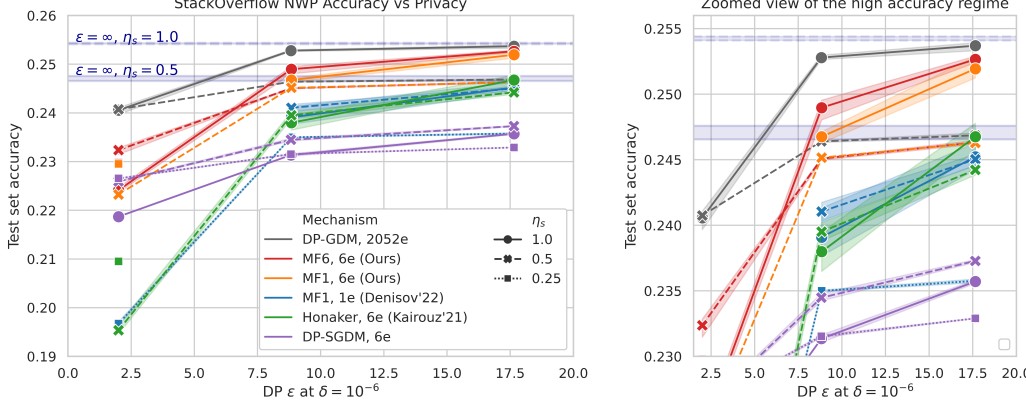

Figure 12: Complete data used in Fig. 3 showing results for server learning rates $\eta_s = 0.5$ and $1.0$, as well as $0.25$ when $\varepsilon{=}2$. All algorithms use momentum $0.95$ and a client learning rate of $1.0$. For the 1 and 6 epoch runs, we observe MF generally tolerates larger learning rates, though lower learning rates perform better for all algorithms at $\varepsilon{=}2$.

### F.4 COMPLETE RESULTS

In this section we give additional details on our main grid of experiments. Fig. 12 uses the same data as Fig. 3, but shows results for each learning rate individually. Tables 4 and 5 give the mean, minimum, maximum, and standard deviation of test-set accuracy corresponding to Fig. 12, as well as the number of replicated experiments ('count').

Table 3 gives the noise multipliers to achieve our various privacy targets $\varepsilon$. Due to a change in the accountant used, we have slightly different $\varepsilon$ targets around 8.8 for the different methods. Note the noise multipliers here are incomparable between the MF and (S)GD mechanisms in terms of the total noise introduced. For matrix factorization, we sample $\mathbf{Z} \sim \mathcal{N}(0, \zeta^2 z^2)$ for noise multiplier $z$; this noise is applied after mapping the raw gradients/updates $\mathbf{x}$ through the linear map $\mathbf{C}$ which is normalized so the total sensitivity is $\zeta$. For the (S)GD mechanisms, we add noise $\mathcal{N}(0, \zeta^2 z^2)$ independently to each model update. In all cases, noise is added to the sum of per-user updates, so the effective noise in the average update scales down with the number of clients per round.

| | | Noise multiplier $z$ | |
| target $\varepsilon$ | MF | DP-SGD (6e) | GD (2052e) |
| --- | --- | --- | --- |
| 17.648 | 0.341 | 0.402 | 15.518 |
| 8.841 | 0.600 | - | - |
| 8.824 | - | 0.491 | 27.493 |
| 2.000 | 2.231 | 0.757 | 106.023 |

Table 3: Noise multiplier parameters for the StackOverflow experiments to achieve various $\varepsilon$s at $\delta = 10^{-6}$. Privacy was computed using the PLD accountant, see Appendix D.2.

| clients/round | $\varepsilon$ | $\eta_s$ | Test set accuracy | | | | |
| --- | --- | --- | --- | --- | --- | --- | --- |
| | | | mean | std | min | max | count |
| 1000 | 2.00 | 0.25 | 0.22654 | 0.00009 | 0.22648 | 0.22660 | 2 |
| | | 0.50 | 0.22592 | 0.00013 | 0.22581 | 0.22606 | 3 |
| | | 1.00 | 0.21869 | | 0.21869 | 0.21869 | 1 |
| | 8.82 | 0.25 | 0.23154 | 0.00015 | 0.23143 | 0.23164 | 2 |
| | | 0.50 | 0.23447 | 0.00033 | 0.23419 | 0.23495 | 4 |
| | | 1.00 | 0.23135 | 0.00041 | 0.23106 | 0.23164 | 2 |
| | 17.65 | 0.25 | 0.23290 | 0.00002 | 0.23289 | 0.23291 | 2 |
| | | 0.50 | 0.23728 | 0.00014 | 0.23710 | 0.23741 | 4 |
| | | 1.00 | 0.23571 | 0.00019 | 0.23558 | 0.23585 | 2 |
| 342477 | 2.00 | 0.50 | 0.24074 | | 0.24074 | 0.24074 | 1 |
| | | 1.00 | 0.24056 | 0.00097 | 0.23886 | 0.24125 | 5 |
| | 8.82 | 0.50 | 0.24640 | 0.00011 | 0.24632 | 0.24647 | 2 |
| | | 1.00 | 0.25279 | 0.00019 | 0.25261 | 0.25306 | 4 |
| | 17.65 | 0.50 | 0.24686 | 0.00025 | 0.24668 | 0.24704 | 2 |
| | | 1.00 | 0.25370 | 0.00039 | 0.25312 | 0.25394 | 4 |

Table 4: Test set accuracy statistics for **DP-SGDM, 6e** (1000 clients/round) and **DP-GDM, 2052e** (342,477 clients/round). Accuracy for **DP-GDM, 2052e** was estimated with 1000 clients/round with an appropriately scaled noise multiplier. The `count` columns gives the number of repeated trials of the given configuration, with $\eta_s$ indicating the server learning rate.

| | $\varepsilon$ | $\eta_s$ | Test set accuracy | | | | |
| --- | --- | --- | --- | --- | --- | --- | --- |
| | | | mean | std | min | max | count |
| Honaker, 6e (Kairouz'21) | 2.00 | 0.25 | 0.20951 | 0.00134 | 0.20857 | 0.21046 | 2 |
| | | 0.50 | 0.19535 | 0.00114 | 0.19429 | 0.19655 | 3 |
| | | 1.00 | 0.00011 | | 0.00011 | 0.00011 | 1 |
| | 8.84 | 0.50 | 0.23952 | 0.00101 | 0.23881 | 0.24023 | 2 |
| | | 1.00 | 0.23799 | 0.00212 | 0.23649 | 0.23949 | 2 |
| | 17.65 | 0.50 | 0.24422 | 0.00037 | 0.24383 | 0.24456 | 3 |
| | | 1.00 | 0.24675 | 0.00128 | 0.24540 | 0.24810 | 5 |
| MF1, 1e (Denisov'22) | 2.00 | 0.25 | 0.19674 | 0.00057 | 0.19633 | 0.19715 | 2 |
| | | 0.50 | 0.00093 | 0.00108 | 0.00017 | 0.00169 | 2 |
| | 8.84 | 0.25 | 0.23500 | 0.00009 | 0.23493 | 0.23506 | 2 |
| | | 0.50 | 0.24105 | 0.00083 | 0.24019 | 0.24190 | 4 |
| | | 1.00 | 0.23909 | 0.00196 | 0.23767 | 0.24132 | 3 |
| | 17.65 | 0.25 | 0.23576 | 0.00019 | 0.23562 | 0.23590 | 2 |
| | | 0.50 | 0.24503 | 0.00056 | 0.24408 | 0.24565 | 6 |
| | | 1.00 | 0.24522 | 0.00102 | 0.24424 | 0.24628 | 3 |
| MF1, 6e (Ours) | 2.00 | 0.25 | 0.22953 | 0.00046 | 0.22921 | 0.22985 | 2 |
| | | 0.50 | 0.22324 | 0.00017 | 0.22308 | 0.22341 | 3 |
| | | 1.00 | 0.00010 | | 0.00010 | 0.00010 | 1 |
| | 8.84 | 0.50 | 0.24516 | | 0.24516 | 0.24516 | 1 |
| | | 1.00 | 0.24676 | 0.00044 | 0.24628 | 0.24714 | 3 |
| | 17.65 | 0.50 | 0.24628 | | 0.24628 | 0.24628 | 1 |
| | | 1.00 | 0.25194 | 0.00085 | 0.25124 | 0.25288 | 3 |
| MF6, 6e (Ours) | 2.00 | 0.25 | 0.22978 | 0.00038 | 0.22939 | 0.23014 | 3 |
| | | 0.50 | 0.23237 | 0.00078 | 0.23147 | 0.23286 | 3 |
| | | 1.00 | 0.22413 | 0.00055 | 0.22365 | 0.22472 | 3 |
| | 8.84 | 0.50 | 0.24509 | 0.00010 | 0.24497 | 0.24515 | 3 |
| | | 1.00 | 0.24897 | 0.00081 | 0.24804 | 0.24955 | 3 |
| | 17.65 | 0.50 | 0.24632 | 0.00015 | 0.24618 | 0.24648 | 3 |
| | | 1.00 | 0.25265 | 0.00025 | 0.25240 | 0.25291 | 3 |

Table 5: Test set accuracy for matrix-factorization based mechanisms. Note: Some 6 epoch runs were conducted with shuffling between epochs, and some were conducted using a fixed order for all epochs (as required by our DP analysis). We saw no impact of reshuffling on the final test set accuracy, and so include all runs in these results regardless of the shuffling setting.

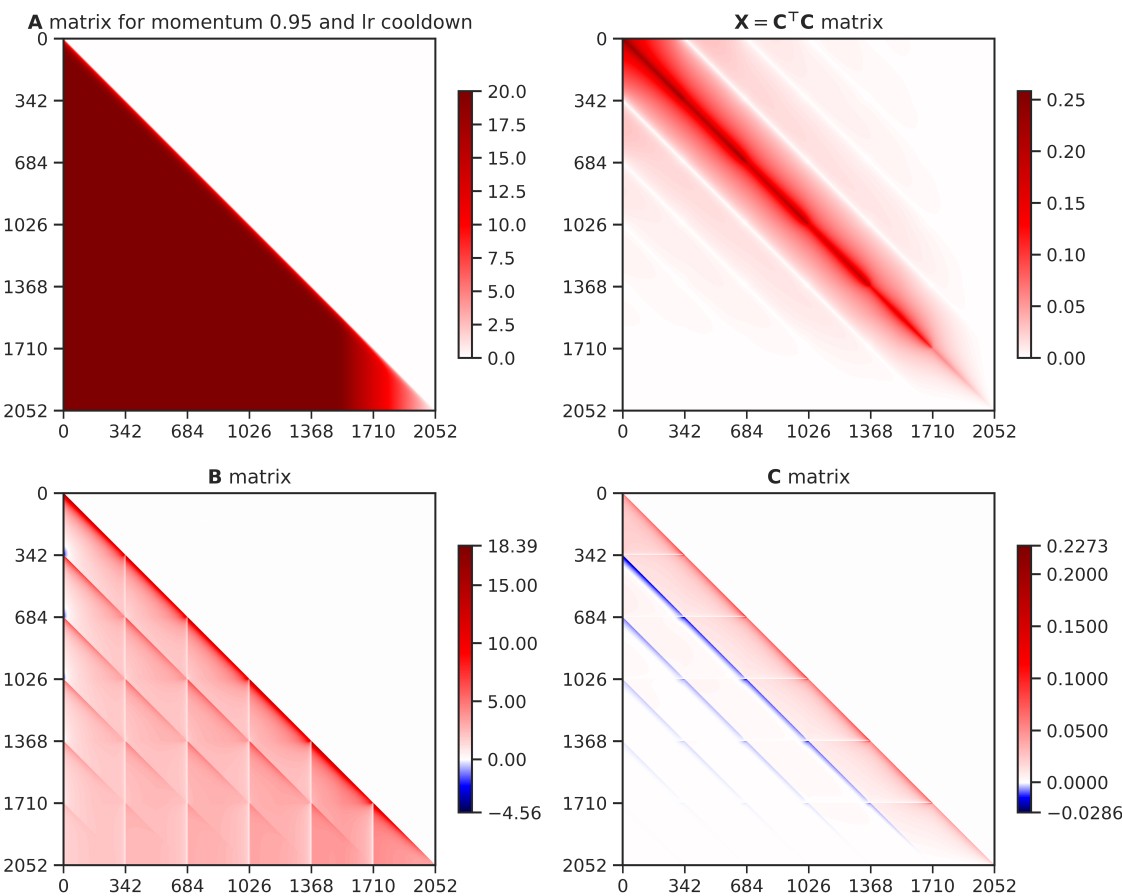

Figure 13: A more detailed view of the matrices shown in Fig. 2.

## F.5 Optimal matrix mechanisms

## G Discussions and Conclusions

Our work significantly improves the privacy-utility tradeoffs in DP ML. Indeed, our work outperforms the state-of-the-art (and, DPSGD with amplification) by $\approx 5$ percentage points across many privacy levels—and as low as $\varepsilon \approx 2$—with practically implementable assumptions. We remark that we compare our mechanisms with DPSGD on a level-ground using well-performing but not state-of-the-art models and training protocols—e.g., very large models, augmentations prior to clipping (De et al., 2022), public data usage, and large batch sizes can all aid training. Many, if not all, of these techniques are applicable to our setting and can thus be used with our mechanisms to realize additional absolute performance gains, again, likely beyond the performances achieved by DPSGD.

The major limitation of our approach is the computation required to generate the optimal matrices. Though our optimal FFT decoder bridges the gap between the mechanisms without optimizer costs and our optimal mechanism, it still leaves some room for improvement in the privacy-utility tradeoff. We believe this is an important area for future work.

## H Analysis for Section 2

### H.1 From scalar to vector contributions

**Theorem H.1.** *Let* $\mathbf{C} \in \mathbb{R}^{n \times k}$ *which satisfies*

$$\|\mathbf{Cu}\|_2 \leq 1 \qquad \forall \mathbf{u} \in \{-1, 1\}^k,$$

*and let $\mathbf{G} \in \mathbb{R}^{k \times d}$ such that each row $\mathbf{G}_{[i,:]}$ for $i \in [k]$ satisfies $\|\mathbf{G}_{[i,:]}\|_2 \leq 1$. Suppose at least one of the following conditions hold:*

1. *We have $k = 1$ or $k = 2$ participations.*

2. *All the entries of $\mathbf{C}^\top \mathbf{C}$ are non-negative.*

3. *The rows of $\mathbf{G}$ are all co-linear, $\mathbf{G}_{[i,:]} = u_i \cdot \mathbf{G}_{[1,:]}$ for $u_i \in \{-1, 1\}$, $i > 1$.*

4. *The rows of $\mathbf{G}$ are all orthogonal, $\langle \mathbf{G}_{[i,:]}, \mathbf{G}_{[j,:]} \rangle = 0$, $\forall i \neq j$, $i, j \in [k]$.*

*Then,*

$$\|\mathbf{CG}\|_F \leq 1. \tag{10}$$

*Furthermore, the following statements are also true without assuming conditions (1)-(3) above.*

- $\|\mathbf{CG}\|_F = \mathcal{O}(\log k)$.

- *If we replace the condition on $\mathbf{G}$ to $\forall i \in [k], \|\mathbf{G}_{[i,:]}\|_1 \leq 1$, then $\|\mathbf{CG}\|_F \leq 1$.*

**Note** Theorem H.1 is generally applied to $\mathbf{C}_{[:,\pi]}$, the sub-matrix of some $\mathbf{C} \in \mathbb{R}^{n \times n}$ formed by keeping only columns selected by a particular participation pattern $\pi$.

*Proof Theorem H.1.* Let $\mathbf{C} = [\mathbf{c}_1 \quad \mathbf{c}_2 \quad \cdots \quad \mathbf{c}_k]$ with each $\mathbf{c}_i \in \mathbb{R}^n$ being a column vector. Also let we write $g_{[i,j]}$ for the $(i, j)$-th entry of $\mathbf{G}$. It will also be useful to note

$$\|\mathbf{CG}\|_F^2 = \mathrm{tr}\big(\mathbf{CG}(\mathbf{CG})^\top\big) = \mathrm{tr}\big(\mathbf{C}^\top \mathbf{CGG}^\top\big). \tag{11}$$

In the following we prove each of the individual cases of Theorem H.1.

**When $k = 1$ or $k = 2$:** For $k = 1$, we have

$$\|\mathbf{CG}\|_F^2 = \|\mathbf{c}_1\|_2^2 \left( \sum_{j=1}^d g_{[1,j]} \right)^2 \leq \|\mathbf{c}_1\|_2^2 = \max_{u \in \{\pm 1\}} \|u \cdot \mathbf{c}_1\|_2^2.$$

An equivalent argument is used in Denisov et al. (2022, Thm. 3.1).

For $k = 2$, we have the following:

$$\|\mathbf{CG}\|_F^2 = \sum_{i=1}^2 \|\mathbf{c}_i\|_2^2 \cdot \left( \sum_{j=1}^d g_{[i,j]}^2 \right) + \big(2g_{[1,1]}g_{[2,1]}\langle \mathbf{c}_1, \mathbf{c}_2 \rangle\big) + \cdots + \big(2g_{[1,d]}g_{[2,d]}\langle \mathbf{c}_1, \mathbf{c}_2 \rangle\big)$$

$$\leq \sum_{i=1}^2 \|\mathbf{c}_i\|_2^2 + \big(2|g_{[1,1]}| \cdot |g_{[2,1]}||\langle \mathbf{c}_1, \mathbf{c}_2 \rangle|\big) + \cdots + \big(2|g_{[1,d]}| \cdot |g_{[2,d]}||\langle \mathbf{c}_1, \mathbf{c}_2 \rangle|\big)$$

$$\leq \sum_{i=1}^2 \|\mathbf{c}_i\|_2^2 + \big(g_{[1,1]}^2 + g_{[2,1]}^2\big) |\langle \mathbf{c}_1, \mathbf{c}_2 \rangle| + \cdots + \big(g_{[1,d]}^2 + g_{[2,d]}^2\big) |\langle \mathbf{c}_1, \mathbf{c}_2 \rangle| \tag{12}$$

$$= \|\mathbf{c}_1\|_2^2 + \|\mathbf{c}_2\|_2^2 + 2|\langle \mathbf{c}_1, \mathbf{c}_2 \rangle| = \max \left\{ (\mathbf{c}_1 + \mathbf{c}_2)^2, (\mathbf{c}_1 - \mathbf{c}_2)^2 \right\}$$

$$\leq \max_{\mathbf{u} \in \{\pm 1\}^2} \|\mathbf{Cu}\|_2^2 \leq 1,$$

where Eq. (12) follows from the standard `A.M.` $\geq$ `G.M.` inequality.

**All the entries of $\mathbf{C}^\top \mathbf{C}$ are non-negative:** Let $\mathbf{X} = \mathbf{C}^\top \mathbf{C}$ and $\hat{\mathbf{G}} = \mathbf{GG}^\top$. Observe $\hat{\mathbf{G}}_{[i,j]} \in [-1, 1]$, and using Eq. (11), when $\mathbf{X}$ is elementwise non-negative, $\mathrm{tr}\big(\mathbf{X}\hat{\mathbf{G}}\big)$ is maximized when $\hat{\mathbf{G}} = \mathbf{1}^{k \times k} = \hat{\mathbf{u}}\hat{\mathbf{u}}^\top$ by choosing $\hat{\mathbf{u}} = \mathbf{1}^k$. Hence,

$$\|\mathbf{CG}\|_F \leq \mathrm{tr}\big(\mathbf{C}^\top \mathbf{C}\hat{\mathbf{u}}\hat{\mathbf{u}}^\top\big) = \|\mathbf{C}\hat{\mathbf{u}}\|_2 \leq \max_{\mathbf{u} \in \{\pm 1\}^k} \|\mathbf{Cu}\|_2^2. \tag{13}$$

Eq. (13) completes the proof.

**The rows of G are co-linear:** By the convexity of $\|\mathbf{CG}\|_F^2$ with respect to the matrix $\mathbf{G}$, we may assume the rows of $\mathbf{G}$ are of $\ell_2$ norm 1. Under the colinearity assumption, this translates to $\mathbf{G}_{[i,:]} = u_i \mathbf{G}_{[1,:]}$, with each $u_i \in \{\pm 1\}$. Let $\mathbf{u} = [u_1, \ldots, u_k] \in \{-1, 1\}^k$. Then for the matrix $\mathbf{GG}^\top$ we have the following:

$$[\mathbf{GG}^\top]_{[i,j]} = \langle \mathbf{G}_{[i,:]}, \mathbf{G}_{[j,:]} \rangle = u_i u_j \langle \mathbf{G}_{[1,:]}, \mathbf{G}_{[1,:]} \rangle = u_i u_j.$$

which implies

$$\mathbf{GG}^\top = \mathbf{uu}^\top. \tag{14}$$

Using Eq. (14) with Eq. (11), we have

$$\|\mathbf{CG}\|_F^2 = \mathrm{tr}\big(\mathbf{CG}(\mathbf{CG})^\top\big) = \mathrm{tr}\big(\mathbf{C}^\top \mathbf{Cuu}^\top\big) \leq \max_{\mathbf{u} \in \{\pm 1\}^k} \|\mathbf{Cu}\|_2^2 \leq 1.$$

**The rows of G are all orthogonal** This condition implies $\hat{\mathbf{G}} = \mathbf{GG}^\top$ is a diagonal matrix with diagonal entries in $[0, 1]$, and so Eq. (11) implies $\|\mathbf{CG}\|_F^2 \leq \mathrm{tr}(\mathbf{X})$. It is thus sufficient to show

$$\mathrm{tr}(\mathbf{X}) \leq \max_{\mathbf{u} \in \{-1,1\}^k} \mathrm{tr}(\mathbf{Xuu}^\top).$$

We give a construction for a $\mathbf{u}$ that shows this. Observe

$$\mathrm{tr}(\mathbf{Xuu}^\top) = \mathrm{tr}(\mathbf{X}) + 2 \sum_{i=1}^{k} u_i \underbrace{\sum_{j=1}^{i-1} u_j \mathbf{X}_{[i,j]}}_{b_i}.$$

Observe we can choose $u_1 = 1$ and then $u_i = \mathrm{sign}(b_i)$ since $b_i$ depends only on $\mathbf{C}$ and the previously fixed $u_j$ for $j < i$, ensuring the double sum on the right is non-negative, completing the proof.

$\|\mathbf{CG}\|_F = \mathcal{O}(\log k)$ **without assuming conditions (1)-(4):** We will prove this claim via probabilistic argument. First notice that due to convexity, we have the following:

$$\max_{\mathbf{u} \in \{\pm 1\}^k} \|\mathbf{Cu}\|_2^2 \leq 1$$

$$\Rightarrow \forall \mathbf{x} \in \mathbb{R}^k, \|\mathbf{Cx}\|_2^2 \leq \|\mathbf{x}\|_\infty^2. \tag{15}$$

We now observe the following for Normal distributions:

$$\|\mathbf{CG}\|_F^2 = \mathbb{E}_{\mathbf{z} \sim \mathcal{N}(0,1)^d} \left[ \|\mathbf{CGz}\|_2^2 \right]$$

$$\leq \mathbb{E}_{\mathbf{z} \sim \mathcal{N}(0,1)^d} \left[ \|\mathbf{Gz}\|_\infty^2 \right] = \mathcal{O}\left( \max_{i \in [k]} \|\mathbf{G}_{[i,:]}\|_2^2 \cdot \log k \right) = \mathcal{O}(\log k). \tag{16}$$

The first inequality in Eq. (16) follows from Eq. (15), and the first equality in Eq. (16) follows from expectation of the maximum of Gaussian random variables.

**Replacing the condition on G to** $\forall i \in [k], \|\mathbf{G}_{[i,:]}\|_1 \leq 1$, **then** $\|\mathbf{CG}\|_F \leq 1$: First notice that since $\|\mathbf{CG}\|_F^2$ is a convex function, the maximum happens at the extreme points of the constraint set $\mathcal{G} = \{\mathbf{G} \mid \forall i \in [k], \|\mathbf{G}_{[i,:]}\|_1 = 1\}$. We use Claim H.1 to identify the extreme points of $\mathcal{G}$.

**Claim H.1** (Theorem 1 in Cao et al. (2022)). *The set of extreme points of the set of $k \times d$ row-stochastic matrices are precisely the set of row permutation matrices, i.e., set of the matrices with entries in $\{0, 1\}^{k \times d}$ and each row has exactly one non-zero entry.*

Notice that the constraint on any matrix $\mathbf{G} \in \mathcal{G}$ is oblivious to the sign, meaning, we can flip the sign of any set of entries in $\mathbf{G}$ and the new matrix will still be in $\mathcal{G}$. This along with Claim H.1 immediately implies that the set $\mathcal{H} = \{H \in \{-1, 0, 1\}^{k \times d} : \forall i \in [k], \|\mathbf{H}_{[i,:]}\|_0 = \|\mathbf{H}_{[i,:]}\|_\infty = 1\}$ is the set of extreme points of the set $\mathcal{G}$. (If the set of extreme points of $\mathcal{G}$ is larger than $\mathcal{H}$, then the signs of any such extreme point can be flipped to create a new extreme point of row-stochastic matrices, which would violate Claim H.1.)

It is not hard to observe that for any $H \in \mathcal{H}$, there exists an $\mathbf{u}_H \{\pm 1\}^k$ s.t. $\|\mathbf{CH}\|_F = \|\mathbf{Cu}_H\|_2$. Since, $\|\mathbf{Cu}_H\|_2 \leq \max_{\mathbf{u} \in \{\pm 1\}^k} \|\mathbf{Cu}\|_2$ for any choice of $\mathbf{u}_H$, and the fact that $\max_{\mathbf{G}} \|\mathbf{CG}\|_F$ is reached at one of the matrices in $\mathcal{H}$, the claim in Theorem H.1 follows. $\qquad \square$

## H.2 A COUNTEREXAMPLE FOR GENERAL $\mathbf{C}$

Theorem H.1 indicates the possibility of the following conjecture being true, because of it being true in so many special cases: If $\max_{\mathbf{u} \in \{\pm 1\}^k} \|\mathbf{Cu}\|_2 \leq 1$, then $\|\mathbf{CG}\|_F \leq 1$ for all $\mathbf{G}$ with row $\ell_2$-norm at most one. Unfortunately, we show that the conjecture is not true when $k > 2$, as shown by the following counterexample with $n = k = 3$ and $d = 2$:

$$\mathbf{C} = \frac{1}{\sqrt{24}} \begin{bmatrix} 2 & 1 & 1 \\ 1 & 2 & -1 \\ 1 & -1 & 2 \end{bmatrix} \quad \text{and} \quad \mathbf{G} = \frac{1}{\sqrt{5}} \begin{bmatrix} 2 & 1 \\ 2 & -1 \\ 1 & 2 \end{bmatrix}$$

Direct calculation shows $\max_{\mathbf{u} \in \{\pm 1\}^k} \|\mathbf{Cu}\|_2 = 1$, but $\|\mathbf{CG}\|_F = \sqrt{1.1} \approx 1.049$.

## H.3 PROOF OF COROLLARY 2.1

**Corollary 2.1 Restated.** *When per-step contributions are bounded by $\zeta = 1$, for any participation pattern $\Pi$ and dimensionality $d \geq 1$, when $\mathbf{C}^\top \mathbf{C} \geq 0$ elementwise, we have*

$$\text{sens}_{\mathfrak{D}_\Pi^d}(\mathbf{C}) = \text{sens}_{\mathfrak{D}_\Pi^1}(\mathbf{C}).$$

*Proof.* To see $\text{sens}_{\mathfrak{D}_\Pi^d}(\mathbf{C}) \geq \text{sens}_{\mathfrak{D}_\Pi^1}(\mathbf{C})$, suppose $\mathbf{u}^\star = \arg\max_{u \in \mathfrak{D}_\Pi^1} \|\mathbf{Cu}\|_2$, and observe we can construct a $\mathbf{G}$ such that $\|\mathbf{CG}\|_F = \|\mathbf{Cu}^\star\|_2$ by taking rows $\mathbf{G}_{[i,:]} = (\mathbf{u}_i^\star, 0, \ldots, 0) \in \mathbb{R}^d$ for $i \in [n]$.

For the other direction, for each $\pi \in \Pi$, we apply Theorem H.1 to the matrix $\mathbf{C}_\pi = \mathbf{C}_{[:,\pi]}$, and observe $\mathbf{C}^\top \mathbf{C} \geq 0$ is sufficient to imply $\mathbf{C}_\pi^\top \mathbf{C}_\pi \geq 0$. $\square$

**Note** The condition $\mathbf{C}^\top \mathbf{C} \geq 0$ is sufficient but not in fact necessary for Corollary 2.1 to hold. In particular, for $(k, b)$-participation $\Pi$, the sub-matrices $\mathbf{C}_\pi^\top \mathbf{C}_\pi$ for $\pi \in \Pi$ "touch" only $k^2 b$ entries of the $n^2 = k^2 b^2$ entries of $\mathbf{C}^\top \mathbf{C}$; the other entries of $\mathbf{C}^\top \mathbf{C}$ could in fact be negative. However, we did not need to use this observation for any of the matrices in our experiments.

## H.4 PROOF OF THEOREM 2.1

**Theorem 2.1 Restated.** *Let $\mathbf{C} \in \mathbb{R}^{n \times n}$, and take some participation pattern $\Pi$, with $k = \max_{\pi \in \Pi} |\pi|$ the maximum number of participations. With $\mathbf{C}_{[:,\pi]}$ representing to selecting the columns of the matrix $\mathbf{C}$ indexed by $\pi$ and $\|\cdot\|_2$ the spectral matrix norm, let $\lambda = \max_{\pi \in \Pi} \|\mathbf{C}_{[:,\pi]}\|_2$. Then*

$$\text{sens}_{\mathfrak{D}_\Pi^1}(\mathbf{C}) \leq \lambda \sqrt{k}.$$

*Proof.* By assumption we have $\|\mathbf{u}\| \leq \sqrt{k}$, and so

$$\max_{\mathbf{u} \in \mathfrak{D}} \|\mathbf{Cu}\|_2 \leq \max_{\pi \in \Pi} \max_{\mathbf{u} \in \mathfrak{D}} \|\mathbf{C}[:,\pi]\|_2 \|\mathbf{u}\|_2 \leq \lambda \sqrt{k}.$$

$\square$

## I ANALYSIS FOR SECTION 3

### I.1 PROOF OF THEOREM 3.1

**Theorem 3.1 Restated.** *Let a finite $\mathfrak{D} = \{\mathbf{u}_i\}_{i=1}^k$ be given, and assume that the vectors $\{\mathbf{u}_i\}_{i=1}^k$ span $\mathbb{R}^n$. Assume that $\mathbf{A}$ is full-rank, and for $\mathbf{v} \in \mathbb{R}^k$ define*

$$\mathbf{H}_\mathbf{v} = [\mathbf{u}_1, \ldots, \mathbf{u}_k] \operatorname{diag}(\mathbf{v})^{1/2}, \quad \mathbf{U} = \mathbf{H}_\mathbf{v} \mathbf{H}_\mathbf{v}^\top.$$

*Then, for Lagrange multipliers $\mathbf{v}$ such that the $\mathbf{U}$ is full-rank, the Lagrange dual function $g$ can be expressed in closed form in terms of the Lagrange multipliers:*

$$g(\mathbf{v}) := \inf_{\mathbf{X} \text{ is PD}} L(\mathbf{X}, \mathbf{v}) = 2 \operatorname{tr}\left( \left( \mathbf{U}^{\frac{1}{2}} \mathbf{A}^\top \mathbf{A} \mathbf{U}^{\frac{1}{2}} \right)^{\frac{1}{2}} \right) - \sum_{\mathbf{u} \in \mathfrak{D}} \mathbf{v}_\mathbf{u} \tag{17}$$

*Proof.* Since there is some finite set of vectors $\mathbf{u} \in \mathbb{R}^n$ specifying $\mathfrak{D}$, the supremum in Eq. (5) may be reduced to a maximum over these elements.

Our problem then takes the form:

$$
\begin{aligned}
\inf_{\mathbf{X} \text{ is PD}} \quad & \operatorname{tr}(\mathbf{A}^\top \mathbf{A} \mathbf{X}^{-1}) \\
\text{s.t.} \quad & \mathbf{u}^\top \mathbf{X} \mathbf{u} \leq 1, \ \forall \mathbf{u} \in \mathfrak{D}.
\end{aligned}
\tag{18}
$$

Recall that we have defined $\mathbf{H_v} = [\mathbf{u}_1, \ldots, \mathbf{u}_k] \operatorname{diag}(\mathbf{v})^{\frac{1}{2}}$, and $\mathbf{U} = \mathbf{H_v} \mathbf{H_v}^\top$. Now, note:

$$
\mathbf{U} = \sum_{\mathbf{u}} \mathbf{v_u} \mathbf{u} \mathbf{u}^\top = \mathbf{H} \operatorname{diag}(\mathbf{v}) \mathbf{H}^\top = \mathbf{H_v} \mathbf{H_v}^\top.
\tag{19}
$$

Introducing Lagrange multipliers $\mathbf{v_u} \geq 0$, for the problem Eq. (18) we form the Lagrangian for positive-definite $\mathbf{X}$:

$$
L(\mathbf{X}, \mathbf{v}) = \operatorname{tr}(\mathbf{A}^\top \mathbf{A} \mathbf{X}^{-1}) + \sum_{\mathbf{u}} \mathbf{v_u} \left( \mathbf{u}^\top \mathbf{X} \mathbf{u} - 1 \right)
\tag{20}
$$

$$
= \operatorname{tr}(\mathbf{A}^\top \mathbf{A} \mathbf{X}^{-1}) + \operatorname{tr}\left( \left( \sum_{\mathbf{u}} \mathbf{v_u} \mathbf{u} \mathbf{u}^\top \right) \mathbf{X} \right) - \sum_{\mathbf{u}} \mathbf{v_u}
\tag{21}
$$

$$
= \operatorname{tr}(\mathbf{A}^\top \mathbf{A} \mathbf{X}^{-1}) + \operatorname{tr}(\mathbf{U} \mathbf{X}) - \sum_{\mathbf{u}} \mathbf{v_u}.
\tag{22}
$$

For fixed $\mathbf{v}$, any finite minimizer of $L$ for positive-definite $\mathbf{X}$ must correspond to a zero of this Lagrangian's gradient. We then compute the gradient

$$
\frac{\partial L}{\partial \mathbf{X}} = -\mathbf{X}^{-1} \mathbf{A}^\top \mathbf{A} \mathbf{X}^{-1} + \mathbf{U}.
\tag{23}
$$

$\mathbf{U}$ and $\mathbf{A}$ are full-rank by assumption; therefore Lemma I.2 is applicable, and Eq. (23) has a unique positive-definite zero (and indeed, the infimum in Eq. (18) becomes a minimum):

$$
\mathbf{X} = \mathbf{U}^{-\frac{1}{2}} \left( \mathbf{U}^{\frac{1}{2}} \mathbf{A}^\top \mathbf{A} \mathbf{U}^{\frac{1}{2}} \right)^{\frac{1}{2}} \mathbf{U}^{-\frac{1}{2}}.
\tag{24}
$$

Note that Eq. (23) also immediately implies that if $\mathbf{U}$ is not full-rank, then there is no finite positive-definite minimizer of $L$ in $\mathbf{X}$. Letting $g(\mathbf{v}) = \min_{\mathbf{X}} L(\mathbf{X}, \mathbf{v})$ be the Lagrange dual function and plugging back into Eq. (22), we have

$$
\begin{aligned}
g(\mathbf{v}) &= \min_{\mathbf{X} \text{ is PD}} \operatorname{tr}(\mathbf{A}^\top \mathbf{A} \mathbf{X}^{-1}) + \operatorname{tr}(\mathbf{U} \mathbf{X}) - \sum_{\mathbf{u}} \mathbf{v_u} \\
&= \min_{\mathbf{X} \text{ is PD}} \operatorname{tr}(\mathbf{X} \mathbf{U}) + \operatorname{tr}(\mathbf{U} \mathbf{X}) - \sum_{\mathbf{u}} \mathbf{v_u} && \text{using Eq. (23)} \\
&= \min_{\mathbf{X} \text{ is PD}} 2 \operatorname{tr}(\mathbf{U} \mathbf{X}) - \sum_{\mathbf{u}} \mathbf{v_u} \\
&= 2 \operatorname{tr}\left( \mathbf{U} \mathbf{U}^{-\frac{1}{2}} \left( \mathbf{U}^{\frac{1}{2}} \mathbf{A}^\top \mathbf{A} \mathbf{U}^{\frac{1}{2}} \right)^{\frac{1}{2}} \mathbf{U}^{-\frac{1}{2}} \right) - \sum_{\mathbf{u}} \mathbf{v_u} && \text{using Eq. (24)} \\
&= 2 \operatorname{tr}\left( \left( \mathbf{U}^{\frac{1}{2}} \mathbf{A}^\top \mathbf{A} \mathbf{U}^{\frac{1}{2}} \right)^{\frac{1}{2}} \right) - \sum_{\mathbf{u}} \mathbf{v_u}.
\end{aligned}
$$

$\square$

### I.1.1 PROOF OF COROLLARY 3.1

**Corollary 3.1 Restated.** *In the same setup as Theorem 3.1, a maximizer of the dual $\mathbf{v}^\star$ must satisfy:*

$$
\mathbf{v}^\star = \operatorname{diagpart}\left( \left( \mathbf{H}_{\mathbf{v}^\star}^\top \mathbf{A}^\top \mathbf{A} \mathbf{H}_{\mathbf{v}^\star} \right)^{\frac{1}{2}} \right).
\tag{25}
$$

*Moreover, the optimal value of the problem defined in 6 is $\operatorname{tr}(\mathbf{v}^\star)$.*

*Proof.* As noted in the remark after Theorem 3.1, any optimal setting of the dual variables must be in the interior of a neighborhood in which the representation Eq. (7) is valid. It is therefore permissible to differentiate this representation.

Differentiating, we find:

$$
\begin{aligned}
\frac{\partial}{\partial \mathbf{v}_i} \mathrm{tr}\left( \left(\mathbf{U}^{\frac{1}{2}}\mathbf{A}^\top \mathbf{A}\mathbf{U}^{\frac{1}{2}}\right)^{\frac{1}{2}} \right) &= \frac{1}{2}\mathrm{tr}\left( \mathbf{A}^\top \mathbf{A}\mathbf{U}^{\frac{1}{2}}(\mathbf{U}^{\frac{1}{2}}\mathbf{A}^\top \mathbf{A}\mathbf{U}^{\frac{1}{2}})^{-\frac{1}{2}}\mathbf{U}^{-\frac{1}{2}}\mathbf{u}_i\mathbf{u}_i^\top \right) \\
&= \frac{1}{2}\mathrm{tr}\left( \mathbf{U}^{-\frac{1}{2}}(\mathbf{U}^{\frac{1}{2}}\mathbf{A}^\top \mathbf{A}\mathbf{U}^{\frac{1}{2}})(\mathbf{U}^{\frac{1}{2}}\mathbf{A}^\top \mathbf{A}\mathbf{U}^{\frac{1}{2}})^{-\frac{1}{2}}\mathbf{U}^{-\frac{1}{2}}\mathbf{u}_i\mathbf{u}_i^\top \right) \\
&= \frac{1}{2}\mathrm{tr}\left( \mathbf{U}^{-\frac{1}{2}}(\mathbf{U}^{\frac{1}{2}}\mathbf{A}^\top \mathbf{A}\mathbf{U}^{\frac{1}{2}})^{\frac{1}{2}}\mathbf{U}^{-\frac{1}{2}}\mathbf{u}_i\mathbf{u}_i^\top \right) \\
&= \frac{1}{2}\mathrm{tr}\left( \mathbf{u}_i^\top \mathbf{U}^{-\frac{1}{2}}(\mathbf{U}^{\frac{1}{2}}\mathbf{A}^\top \mathbf{A}\mathbf{U}^{\frac{1}{2}})^{\frac{1}{2}}\mathbf{U}^{-\frac{1}{2}}\mathbf{u}_i \right) \\
&= \frac{1}{2}\mathbf{u}_i^\top \mathbf{X}\mathbf{u}_i,
\end{aligned}
$$

by defining $\mathbf{X} = \mathbf{U}^{-\frac{1}{2}}\left(\mathbf{U}^{\frac{1}{2}}\mathbf{A}^\top \mathbf{A}\mathbf{U}^{\frac{1}{2}}\right)^{\frac{1}{2}}\mathbf{U}^{-\frac{1}{2}}$ (recalling the usage of the symbol $\mathbf{X}$ in Eq. (24)).

Therefore

$$
\frac{\partial g(\mathbf{v})}{\partial \mathbf{v}_i} = \mathbf{u}_i^\top \mathbf{X}\mathbf{u}_i - 1,
$$

and we have the stated expression for the gradient of the dual function.

Now, at a maximizer of the dual function, this derivative must vanish. An equivalent condition is $\mathrm{diagpart}(\mathbf{H}^\top \mathbf{X}\mathbf{H}) = \vec{1}$, and hence

$$
\mathrm{tr}(\mathbf{U}\mathbf{X}) = \mathrm{tr}(\mathbf{H}\,\mathrm{diag}(\mathbf{v})\mathbf{H}^\top \mathbf{X}) = \mathrm{tr}(\mathrm{diag}(\mathbf{v})\mathbf{H}^\top \mathbf{X}\mathbf{H}) = \sum_{\mathbf{u}} \mathbf{v}_{\mathbf{u}}, \tag{26}
$$

so at the optimum $\mathbf{v}^\star$ in fact $g(\mathbf{v}^\star) = \sum_{\mathbf{u}} \mathbf{v}^\star{}_{\mathbf{u}}$, establishing the second claim of our result.

Again using the observation that $\mathrm{diagpart}(\mathbf{H}_\mathbf{v}^\top \mathbf{X}\mathbf{H}_\mathbf{v}) = \vec{1}$ and so

$$
\mathrm{diagpart}(\mathbf{H}_\mathbf{v}^\top \mathbf{X}\mathbf{H}_\mathbf{v}) = \mathrm{diagpart}(\mathrm{diag}(\mathbf{v})\mathbf{H}^\top \mathbf{X}\mathbf{H}) = \mathbf{v}.
$$

Further, using the second claim of Corollary I.1, we can take

$$
\mathbf{X} = \mathbf{H}_\mathbf{v}^{-\top}\left(\mathbf{H}_\mathbf{v}^\top \mathbf{A}^\top \mathbf{A}\mathbf{H}_\mathbf{v}\right)^{\frac{1}{2}}\mathbf{H}_\mathbf{v}^{-1},
$$

and multiplying this by $\mathbf{H}_\mathbf{v}^\top$ and $\mathbf{H}_\mathbf{v}$ on the left and right respectively yields

$$
\mathbf{H}_\mathbf{v}^\top \mathbf{X}\mathbf{H}_\mathbf{v} = \mathbf{H}_\mathbf{v}^\top \mathbf{H}_\mathbf{v}^{-\top}\left(\mathbf{H}_\mathbf{v}^\top \mathbf{A}^\top \mathbf{A}\mathbf{H}_\mathbf{v}\right)^{\frac{1}{2}}\mathbf{H}_\mathbf{v}^{-1}\mathbf{H}_\mathbf{v} = \left(\mathbf{H}_\mathbf{v}^\top \mathbf{V}\mathbf{H}_\mathbf{v}\right)^{\frac{1}{2}}
$$

and so we conclude for the optimal Lagrange multiplier $\mathbf{v}^\star$,

$$
\mathbf{v}^\star = \mathrm{diagpart}\left( \left(\mathbf{H}_{\mathbf{v}^\star}^\top \mathbf{A}^\top \mathbf{A}\mathbf{H}_{\mathbf{v}^\star}\right)^{\frac{1}{2}} \right). \tag{27}
$$

$\square$

## I.2 LEMMAS AND COROLLARIES

**Lemma I.1.** *The set of positive-definite $\mathbf{X}$ such that $\sup_{\mathbf{u}\in\mathfrak{D}} \mathbf{u}^\top \mathbf{X}\mathbf{u} \leq 1$ is bounded as a subset of $\mathbb{R}^{n\times n}$ if and only if $\mathfrak{D} = \{\mathbf{u}\}$ spans $\mathbb{R}^n$.*

*Proof.* Suppose that $\mathfrak{D}$ spans $\mathbb{R}^n$. For a PSD matrix, a bound on the trace implies a bound on the elements; therefore it is sufficient to show that $\sup_{\mathbf{u}\in\mathfrak{D}} \mathbf{u}^\top \mathbf{X}\mathbf{u} \leq 1$ implies that the maximum eigenvalue of $\mathbf{X}$ is uniformly bounded for $\mathbf{X}$ PSD.

Take some set of vectors $\{\mathbf{u}_i\}_{i=1}^n \in \mathfrak{D}$ which span $\mathbb{R}^n$. Fix some representation

$$\mathbf{e}_i = \sum_{j=1}^n \alpha_{ij} \mathbf{u}_j,$$

where $\mathbf{e}_i$ is the $i^{th}$ standard basis vector.

Take $\mathbf{y}$ of $\ell_2$ norm 1, and express:

$$\mathbf{y} = \sum_{i=1}^n \gamma_i \mathbf{e}_i.$$

Then for $\mathbf{X}$ satisfying our assumptions,

$$\left| \mathbf{y}^\top \mathbf{X} \mathbf{y} \right| = \left| \sum_{i,j=1}^n \gamma_i \gamma_j \mathbf{e}_i^\top \mathbf{X} \mathbf{e}_j \right| = \leq n^2 \max_{1 \leq i,j \leq n} |\gamma_i \gamma_j| \left| \mathbf{e}_i^\top \mathbf{X} \mathbf{e}_j \right|. \tag{28}$$

Similarly,

$$\left| \mathbf{e}_i^\top \mathbf{X} \mathbf{e}_j \right| = \left| \sum_{k,l=1}^n \alpha_{ik} \alpha_{jl} \mathbf{u}_k^\top \mathbf{X} \mathbf{u}_l \right| \leq n^2 \max_{1 \leq k,l \leq n} |\alpha_{ik} \alpha_{jl}| \left| \mathbf{u}_k^\top \mathbf{X} \mathbf{u}_l \right| \leq n^2 \max_{1 \leq k,l \leq n} |\alpha_{ik} \alpha_{jl}|, \tag{29}$$

where the final inequality follows by the assumptions on $\mathbf{X}$.

Now, since the $\ell_2$ norm of $\mathbf{y}$ is 1, the orthogonality of the $\mathbf{e}_i$ imply that each $|\gamma_i \gamma_j|$ is at most 1. Therefore:

$$\left| \mathbf{y}^\top \mathbf{X} \mathbf{y} \right| \leq n^4 \max_{1 \leq i,j,k,l \leq n} |\alpha_{ik} \alpha_{jl}|, \tag{30}$$

and we have sufficiency of $\mathfrak{D}$ spanning $\mathbb{R}^n$.

For necessity, suppose $\mathfrak{D}$ does not span $\mathbb{R}^n$. Then there is some vector $\mathbf{y} \in \text{span}(\mathfrak{D})^\perp$ of norm 1. Take any $\mathbf{X}$ such that $\sup_{\mathbf{u} \in \mathfrak{D}} \mathbf{u}^\top \mathbf{X} \mathbf{u} \leq 1$. Then, since $\mathbf{y}^\top \mathbf{u} = 0$ for all $\mathbf{u} \in \mathfrak{D}$, $\mathbf{Y} := \mathbf{X} + \alpha \mathbf{y}$ satisfies the same set of inequalities for any $\alpha$. $\quad\square$

**Lemma I.2.** *Let* $\mathbf{U}, \mathbf{V} \in S_{++}^n$. *Let* $\mathbf{U} = \mathbf{U}_L \mathbf{U}_R$ *be a factorization of* $\mathbf{U}$ *such that* $\mathbf{U}_R \mathbf{V} \mathbf{U}_L$ *is PSD, and the following equation defines a positive-definite matrix* $\mathbf{X}$:

$$\mathbf{X} = \mathbf{U}_R^\dagger \left( \mathbf{U}_R \mathbf{V} \mathbf{U}_L \right)^{\frac{1}{2}} \mathbf{U}_L^\dagger. \tag{31}$$

*Then, this* $\mathbf{X}$ *solves the equation*

$$\mathbf{X} \mathbf{U} \mathbf{X} = \mathbf{V} \quad \text{or equivalently} \quad \mathbf{U} = \mathbf{X}^{-1} \mathbf{V} \mathbf{X}^{-1}.$$

*Moreover, this positive-definite solution* $\mathbf{X}$ *is unique.*

*Proof.* We will begin by showing that $\mathbf{X}$ as defined by Eq. (31) solves the equation $\mathbf{X} \mathbf{U} \mathbf{X} = \mathbf{V}$; then we will show that any two positive-definite representations of the form Eq. (31) are in fact identical.

Notice that the representation $\mathbf{U} = \mathbf{U}_L \mathbf{U}_R$ implies that $\text{rank}(\mathbf{U}_L) \geq n$ and $\text{rank}(\mathbf{U}_R) \geq n$. Therefore $\mathbf{U}_L \mathbf{U}_L^\dagger = \mathbf{I} = \mathbf{U}_R^\dagger \mathbf{U}_R$, as implied by the Moore definition of the Moore-Penrose pseudoinverse. So:

$$
\begin{aligned}
\mathbf{XUX} &= \mathbf{XU}_L\mathbf{U}_R\mathbf{X} \\
&= \left(\mathbf{U}_R^\dagger(\mathbf{U}_R\mathbf{VU}_L)^{\frac{1}{2}}\mathbf{U}_L^\dagger\right)\mathbf{U}_L\mathbf{U}_R\left(\mathbf{U}_R^\dagger(\mathbf{U}_R\mathbf{VU}_L)^{\frac{1}{2}}\mathbf{U}_L^\dagger\right) \\
&= \mathbf{U}_R^\dagger(\mathbf{U}_R\mathbf{VU}_L)^{\frac{1}{2}}\mathbf{P}_{R(\mathbf{U}_L^\dagger)}\mathbf{P}_{R(\mathbf{U}_R)}(\mathbf{U}_R\mathbf{VU}_L)^{\frac{1}{2}}\mathbf{U}_L^\dagger
\end{aligned}
$$

We claim that $(\mathbf{U}_R\mathbf{VU}_L)^{\frac{1}{2}}\mathbf{P}_{R(\mathbf{U}_L^\dagger)} = \mathbf{P}_{R(\mathbf{U}_R)}(\mathbf{U}_R\mathbf{VU}_L)^{\frac{1}{2}} = (\mathbf{U}_R\mathbf{VU}_L)^{\frac{1}{2}}$. In both cases, this can be seen by multiplying on the left or right as appropriate by $(\mathbf{U}_R\mathbf{VU}_L)^{\frac{1}{2}}$, and noting $\mathbf{P}_{R(\mathbf{U}_R)}\mathbf{U}_R = \mathbf{U}_R$, $\mathbf{U}_L\mathbf{P}_{R(\mathbf{U}_L^\dagger)} = \mathbf{U}_L$. Since all the terms here are symmetric, the appropriate equality follows by uniqueness of the symmetric matrix square root. Therefore:

$$
\begin{aligned}
\mathbf{XUX} &= \mathbf{U}_R^\dagger\mathbf{U}_R\mathbf{VU}_L\mathbf{U}_L^\dagger \\
&= \mathbf{V}.
\end{aligned}
$$

The uniqueness of a positive-definite $\mathbf{X}$ solving $\mathbf{XUX} = \mathbf{V}$ follows from the uniqueness of the usual matrix square root. Indeed, assume $\mathbf{Y}$ positive-definite satisfies $\mathbf{YUY} = \mathbf{V}$. Then:

$$
\left(\mathbf{U}^{1/2}\mathbf{YU}^{1/2}\right)^2 = \mathbf{U}^{1/2}\mathbf{VU}^{1/2}
$$

Since the positive-definite square root is uniquely determined, $\mathbf{U}^{1/2}\mathbf{YU}^{1/2}$ is uniquely determined. Since $\mathbf{U}$ is invertible, $\mathbf{Y}$ is uniquely determined as well, and we have $\mathbf{Y} = \mathbf{X}$. $\qquad\square$

**Corollary I.1.** *Two particular instantiations of Lemma I.2 are of interest.* $\mathbf{X}$ *as the matrix geometric mean of* $\mathbf{U}^{-1}$ *and* $\mathbf{V}$ *(taking* $\mathbf{U}_L = \mathbf{U}_R = \sqrt{\mathbf{U}}$*):*

$$
\mathbf{X} = \mathbf{U}^{-\frac{1}{2}}\left(\mathbf{U}^{\frac{1}{2}}\mathbf{VU}^{\frac{1}{2}}\right)^{\frac{1}{2}}\mathbf{U}^{-\frac{1}{2}}, \tag{32}
$$

*and assuming the representation* $\mathbf{U} = \mathbf{H}_\mathbf{v}\mathbf{H}_\mathbf{v}^\top$*:*

$$
\mathbf{X} = \mathbf{H}_\mathbf{v}^{\dagger\top}\left(\mathbf{H}_\mathbf{v}^\top\mathbf{VH}_\mathbf{v}\right)^{\frac{1}{2}}\mathbf{H}_\mathbf{v}^\dagger. \tag{33}
$$

*Proof.* By positive-definiteness of $\mathbf{U}$ and $\mathbf{V}$, Eq. (32) is clearly positive definite; Eq. (33) may be seen to be positive definite via the SVD of the pseudoinverses involved. Symmetry is again clear. Therefore both representations satisfy the assumptions of Lemma I.2. $\qquad\square$

## J   ANALYSIS FOR SECTION 4

### J.1   ADDITIONAL DETAILS

**Defining the circulant matrix**   We consider the special case where $\mathbf{A}$ is the prefix sum linear query matrix (lower-triangle matrix of ones). Then, we define the corresponding circulant matrix

$$
\mathbf{A}_{\mathsf{circ}} \triangleq \begin{bmatrix} v_0 & v_{2n-1} & \cdots & v_1 \\ v_1 & v_0 & \cdots & v_2 \\ \vdots & \vdots & \cdots & \vdots \\ v_{2n-1} & v_{2n-2} & \cdots & v_0 \end{bmatrix} \quad \text{where} \quad \mathbf{v} \triangleq [\underbrace{1,\ldots,1}_{n}, \underbrace{0,\ldots,0}_{n}]. \tag{34}
$$

It is straightforward to verify $\mathbf{A}_{\mathsf{circ}[:n,:n]} = \mathbf{A}$.

**Defining the DFT**

$$
\forall k \in [2n-1]: \mathbf{v}^{\mathsf{DFT}}[k] = \sum_{a=0}^{2n-1} \mathbf{v}(a)\exp\left(\frac{-j2\pi ka}{2n}\right) \tag{35}
$$

**Circulant matrices expressed using Fourier Transforms**

**Theorem J.1** (Adapted from Gray (2006)). *Consider any circulant matrix* $\mathbf{A}_{\text{circ}} \in \mathbb{R}^{2n \times 2n}$. *Let* $\mathbf{F} \in \mathbb{C}^{2n \times 2n}$, *where the k-th row of* $\mathbf{F}$ *is given by* $\mathbf{F}[k,:] = \frac{1}{\sqrt{2n}} \left[ \exp\left( -\frac{j 2\pi k a}{2n} \right) : a \in \{0, \ldots, 2n-1\} \right]$. *Then,* $\mathbf{A}_{\text{circ}} = \mathbf{F}^* \Sigma \mathbf{F}$, *where* $\Sigma \in \mathbb{C}^{2n \times 2n}$ *is a diagonal matrix with the diagonal being the DFT (defined in Equation 35) of the first column of* $\mathbf{A}_{\text{circ}}$. *Here,* $^*$ *is the Hermimitian operation.*

**Privacy and utility guarantees**  In the following we provide the privacy guarantee and the main utility guarantee for the FFT mechanism defined in Algorithm 1.

**Theorem J.2** (DP-Prefix Sum via FFT Privacy Guarantee). *Algorithm 1 is ρ-zCDP in the adaptive continuous release model.*

Next, we analyze the utility of Algorithm 1 and show that it is nearly optimal in terms of the mean squared error (MSE) in the single-pass setting. First, we express the MSE in Theorem J.3 below.

**Theorem J.3** (DP-Prefix Sum via DFT Utility). *The MSE achieved by Algorithm 1 using the real and imaginary components of* $\widetilde{\mathbf{z}}$ *is*

$$\mathbb{E}\left[ MSE \right] = \frac{\kappa^2 \left\| \mathbf{v}^{\mathsf{DFT}} \right\|_1^2}{2\rho n^2}.$$

In the following, we will have an explict expression for $\left\| \mathbf{v}^{\mathsf{DFT}} \right\|_1$ in terms of the problem parameters. Finally, we will argue that Theorem J.3 is nearly optimal.

**Corollary J.1.** *The expected mean squared error (MSE) is given by the following:*

$$\mathbb{E}\left[ MSE \right] = \frac{\kappa^2}{2\rho n^2} \left( n + \sum_{a=0}^{\lfloor \frac{2n-1}{2} \rfloor} \frac{1}{\sin\left( \frac{\pi(2a+1)}{2n} \right)} \right)^2.$$

**Near-optimal utility**  Here, we show that Theorem J.3 is near-optimal in utility for the single-participation setting. To do this, we compare with a lower bound on the expected MSE of any factorization-based mechanism from Henzinger et al. (2022, Theorem 2): $\frac{1}{2\rho\pi^2} \left( 2 + \ln\left( \frac{2n+1}{3} \right) + \frac{\ln(2n+1)}{2n} \right)^2$. We find that the though our analytical upper bound in Corollary J.1 is $\approx 6$x worse than the lower bound, the empirical noise added in Algorithm 1 closely tracks the lower bound to within a factor of 1.2x—because it only adds the real part of the noise. Results are in Figure 14 of Appendix J.1.

**Showing near-optimal utility via MSE experiments**

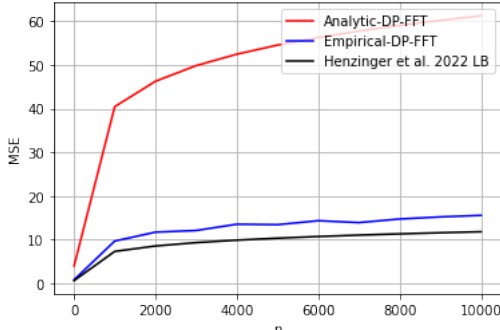

Figure 14: **Algorithm 1 achieves near-optimal utility** as measured by the analytic lower bound from Henzinger et al. (2022, Theorem 2).

J.2    PROOF OF THEOREM J.2

**Theorem J.2 Restated.** *Algorithm 1 is ρ-zCDP in the adaptive continuous release model.*

*Proof.* First, consider the non-adaptive setting and the following mechanism, with parameters as defined in Algorithm 1,

$$\left[\Sigma\left(\mathbf{F}\mathbf{x}_{\mathsf{ext}} + \Sigma^{-1}\mathbf{z}\right)\right],$$

$$\text{where } \mathbf{z} = \sqrt{\frac{\kappa^2 \left\|\mathbf{v}^{\mathsf{DFT}}\right\|_1}{4n\rho}} \left(\sqrt{\Sigma}\cdot\mathbf{w}\right) \tag{36}$$

We claim that this satisfies $\frac{\kappa^2\|\mathbf{v}^{\mathsf{DFT}}\|_1}{4n\sigma^2}$-zCDP. To see this, we proceed by bounding $\rho_i$ for each coordinate $i \in [2n]$ defined in Equation 36. For brevity, let $\mathbf{b} = \mathbf{F}\mathbf{x}_{\mathsf{ext}}$. Consider two neighboring data sets $\mathbf{g}$ and $\mathbf{g}'$, correspondingly, $(\mathbf{b}, \mathbf{x}_{\mathsf{ext}})$ and $(\mathbf{b}', \mathbf{x}'_{\mathsf{ext}})$. Then,

$$\|\mathbf{b} - \mathbf{b}'\|_\infty = \|\mathbf{F}(\mathbf{x}_{\mathsf{ext}} - \mathbf{x}'_{\mathsf{ext}})\|_\infty = \frac{\kappa}{\sqrt{2n}}. \tag{37}$$

We will now prove zCDP guarantee independently for each of the $2n$ coordinates and then use standard zCDP composition (Bun & Steinke, 2016). For any coordinate $a \in \{0, \ldots, 2n - 1\}$, adding noise $\left(\frac{\sigma}{\sqrt{|\mathbf{v}^{\mathsf{DFT}}[i]|}}\right) \cdot \mathcal{N}_{\mathsf{complex}}(0, 1)$ to $\mathbf{b}[i]$ satisfies $\rho_i$-zCDP with $\rho_i = \frac{\kappa^2|\mathbf{v}^{\mathsf{DFT}}[i]|}{4n\sigma^2}$. Then by composition, we have that

$$\rho = \sum_{a=0}^{2n-1}(\rho_i) = \frac{\kappa^2}{4n\sigma^2}\sum_{a=0}^{2n-1}\left(\left|\mathbf{v}^{\mathsf{DFT}}[i]\right|\right) = \frac{\kappa^2\left\|\mathbf{v}^{\mathsf{DFT}}\right\|_1}{4n\sigma^2}. \tag{38}$$

Therefore, setting $\sigma^2 = \frac{\kappa^2\|\mathbf{v}^{\mathsf{DFT}}\|_1}{4n\rho}$-satisfies a non-adaptive $\rho$-zCDP. Using the same $\sigma$, we prove the adaptive part using the same $\sigma$. We have the following from Equation 36.

$$\left[\mathbf{F}^*\left(\Sigma\mathbf{F}\mathbf{x}_{\mathsf{ext}} + \mathbf{z}\right)\right] = \left[\mathbf{F}^*\sqrt{\Sigma}\left(\sqrt{\Sigma}\mathbf{F}\mathbf{x}_{\mathsf{ext}} + \frac{1}{\sqrt{\Sigma}}\mathbf{z}\right)\right] = \left[\mathbf{F}^*\sqrt{\Sigma}\left(\sqrt{\Sigma}\mathbf{F}\mathbf{x}_{\mathsf{ext}} + \sqrt{\frac{\kappa^2\left\|\mathbf{v}^{\mathsf{DFT}}\right\|_1}{4n\rho}}\cdot\mathbf{w}\right)\right] \tag{39}$$

Since, $\mathbf{w}$ in Equation 39 is spherical Gaussian, and the original query matrix $\mathbf{A}$ is lower triangular, by Theorem 2.1 in Denisov et al. (2022), the adaptive privacy guarantee follows. $\square$

## J.3 PROOF OF THEOREM J.3

**Theorem J.3 Restated.** *The MSE achieved by Algorithm 1 using the real and imaginary components of $\widetilde{\mathbf{z}}$ is*

$$\mathbb{E}[MSE] = \frac{\kappa^2\left\|\mathbf{v}^{\mathsf{DFT}}\right\|_1^2}{2\rho n^2}.$$

*Proof.* The MSE is given by the following:

$$\mathbb{E}[\mathrm{MSE}] = \frac{1}{n}\mathbb{E}\left[\|\widetilde{\mathbf{z}}[0, \ldots, n-1]\|_2^2\right] = \frac{\kappa^2\left\|\mathbf{v}^{\mathsf{DFT}}\right\|_1}{2n^2\rho}\cdot\mathrm{tr}\left(|\Sigma[:n,:n]|\right) = \frac{\kappa^2\left\|\mathbf{v}^{\mathsf{DFT}}\right\|_1^2}{2n^2\rho}. \tag{40}$$

In equation 40, $\Sigma[:n,:n]$ refers to the top-left $n \times n$ submatrix of $\Sigma$. $\square$

## J.4 PROOF OF COROLLARY J.1

**Corollary J.1 Restated.** *Under the same setting as Theorem J.3, the MSE for Algorithm 1 is the following*

$$\mathbb{E}[MSE] = \frac{\kappa^2}{2\rho n^2}\left(n + \sum_{a=0}^{\lfloor\frac{2n-1}{2}\rfloor}\frac{1}{\sin\left(\frac{\pi(2a+1)}{2n}\right)}\right)^2.$$

*Proof.* Recall the definition of DFT from Equation 35 and of $\mathbf{v}$ in Equation 34. It is immediate that $\mathbf{v}^{\mathsf{DFT}}[0] = n$. For any $k \neq 0$, we have,

$$\mathbf{v}^{\mathsf{DFT}}[k] = \frac{1 - \exp\left(\frac{-j2\pi kn}{2n}\right)}{1 - \exp\left(\frac{-j2\pi k}{2n}\right)} = \frac{1 - \exp\left(-j\pi k\right)}{1 - \exp\left(\frac{-j\pi k}{n}\right)}. \tag{41}$$

From Equation 41, we have that when $k > 0$ is even, $\mathbf{v}^{\mathsf{DFT}}[k] = 0$. For $k$ odd, we have

$$\left|\mathbf{v}^{\mathsf{DFT}}[k]\right| = \left|\frac{2}{1 - \exp\left(\frac{-j\pi k}{n}\right)}\right| = \frac{1}{|\sin(\pi k/(2n))|} = \frac{1}{\sin(\pi k/(2n))} \tag{42}$$

Combining these, the term $\left\|\mathbf{v}^{\mathsf{DFT}}\right\|_1$ is

$$\left\|\mathbf{v}^{\mathsf{DFT}}\right\|_1 = n + \sum_{a=0}^{\lfloor \frac{n-1}{2} \rfloor} \frac{1}{\sin(\pi(2a+1)/(2n))} \tag{43}$$

$\square$

## J.5 Proof of Theorem 4.1

**Theorem 4.1 Restated.** *Under $k$ participation, Algorithm 1 satisfies $(k^2\rho)$-zCDP.*

*Proof.* The proof goes exactly as Theorem J.2, except equation 37 gets replaced by the following:

$$\|\mathbf{b} - \mathbf{b}'\|_\infty = \|\mathbf{F}(\mathbf{x}_{\mathsf{ext}} - \mathbf{x}'_{\mathsf{ext}})\|_\infty = \frac{k\kappa}{\sqrt{2n}}. \tag{44}$$

$\square$

## K Two related FFT mechanisms.

The FFT mechanism presented in Section 4 can be understood as an application of a complex-valued matrix mechanism factorizing the prefix-sum matrix as

$$\mathbf{B} = \mathbf{PF}^*\sqrt{\Sigma},$$

$$\mathbf{C} = \sqrt{\Sigma}\mathbf{FE},$$

where $\mathbf{E}$ and $\mathbf{P}$ are appropriate embedding and projection matrices, respectively embedding an $n$-dimensional vector in the first $n$ components of $\mathbb{R}^{2n}$, and projecting those same first $n$ components back to $\mathbb{R}^n$, and following this application by 'chopping off' the imaginary part of the noise. The entries of $\Sigma$ may be computed exactly; they contain no purely negative entries, so specifying the principal branch of the square root resolves the implicit ambiguity in the formulation above. This branch corresponds as well to the implementation of the complex square root in major software frameworks (e.g., NumPy).

All these operations are linear; and since everything begins and ends in the real domain, this mechanism can be expressed as a real-valued mechanism. Therefore identical codepaths can be used for implementing experiments with the FFT, though notably without some special implementation of the mechanism, realizing the potential computational savings will not be immediate. In this small section, we translate this complex-valued mechanism into *two* real-valued mechanism which can be integrated with the code backing the rest of the paper. These mechanisms differ in their decoding matrix $\mathbf{B}$, and thus achieve different levels of loss. Both have efficient implementations, though with asymptotics differing by a logarithmic factor. We implement and experiment with *both* of these mechanisms in the main body.

These two mechanisms share an encoding matrix:

$$\mathbf{C_F} = \mathbf{F}^*\mathbf{C},$$

which is real-valued by Lemma K.1. Note that the sensitivity of $\mathbf{C_F}$ is identical to that of $\mathbf{C}$ for any notion of sensitivity expressible as Definition 1 due to the unitary of the Fourier transform. Since this matrix is of shape $[2n, n]$, there is choice in computing the decoder $\mathbf{B}$ such that $\mathbf{BC_F}$ represents the prefix-sum matrix. The two decoders we present below correspond to two subtly distinct mechanisms.

**Mechanism 1: A real-valued version of the mechanism presented in Section 4.** One natural translation of the analysis in Section 4 (indeed, a real-valued version of the precise operation described in Algorithm 1) may be computed by inserting a Fourier transform to match the inverse transform in $\mathbf{C_F}$:

$$\mathbf{B_F} = \mathbf{BF},$$

Clearly $\mathbf{B_F}\mathbf{C_F} = \mathbf{BC}$, and $\mathbf{B_F}$ real-valued by Lemma K.1.

**Proposition K.1.** *For any $\mathfrak{D}$, the mechanism described in Section 4 is distributionally equivalent to an application of the real-valued matrix mechanism with the factorization $(\mathbf{B_F}, \mathbf{C_F})$, and satisfies the same privacy guarantees.*

*Proof.* To show this result, by noting that $\mathbf{C_F}$ and $\mathbf{C}$ have the same sensitivity, it suffices to show that it suffices to show:

- $\Re[\mathbf{F}^*\sqrt{\Sigma}\mathbf{z}]$ (for $\mathbf{z}$ a sample from an isotropic complex Gaussian) is distributionally equivalent to $\mathbf{PF}^*\sqrt{\Sigma}\mathbf{Fb}$ for $\mathbf{b}$ a sample from a real (isotropic) Gaussian with the same variance.

This is a consequence of the distributional invariance of the Gaussian under unitary transformations:

$$\Re[\mathbf{F}^*\sqrt{\Sigma}\mathbf{z}] \sim \Re[\mathbf{F}^*\sqrt{\Sigma}\mathbf{Fz}]$$
$$= \mathbf{F}^*\sqrt{\Sigma}\mathbf{F}\Re[\mathbf{z}] \quad \text{(as } \mathbf{F}^*\sqrt{\Sigma}\mathbf{F} \text{ is real)}$$
$$\sim \mathbf{F}^*\sqrt{\Sigma}\mathbf{Fb},$$

where the variances are as desired. $\square$

Note that the efficiency of the mechanism described in Section 4 carries over immediately to this factorization $(\mathbf{B_F}, \mathbf{C_F})$; indeed, the capacity to compute the noise $\mathbf{B_F}\mathbf{b}$ with complexity $n \log(n)$ may be reasoned to directly, in a similar manner.

This mechanism is not, however, the optimal one for the encoder $\mathbf{C_F}$, and this subtlety has difficult downstream effects in integrating with real-valued factorization codepaths (e.g., see the discussion in Appendix D.4). We proceed to show that the optimal decoder can be used directly, at only a moderate loss of efficiency with sufficiently careful implementation.

**Mechanism 2: A real-valued optimal decoder with complexity $n \log^2(n)$.** As noted in the literature (e.g. Section 3 of Denisov et al. (2022)), for a fixed encoder, the optimal decoder may always be computed in terms of an appropriate pseudoinversion of the encoder. Therefore, we may compute the optimal decoder for the encoder $\mathbf{C_F}$, defining:

$$\mathbf{B_{Fopt}} = \mathbf{SC_F^\dagger},$$

where $\mathbf{S}$ is the prefix-sum matrix. Since $\mathbf{C_F}$ is real, its pseudoinverse is as well, and $\mathbf{B_{Fopt}}$ is also real-valued. Since $\mathbf{B_{Fopt}}$ can have no more variance than $\mathbf{B_F}$, all the utility analysis of the DFT mechanism in Section 4 carries through as an upper bound for this factorization. Privacy of this mechanism is ensured by the fact that this mechanism reuses teh encoder $\mathbf{C_F}$. The major way in which these mechanisms operationally differ comes down to the cost of computing the noise vector $\mathbf{B_{Fopt}}\mathbf{b}$, where $\mathbf{b}$ represents a sample from an isotropic Gaussian distribution. Though we do not know of a complexity result which matches the decoder $\mathbf{B_F}$, we will show that the complexity cost which must be paid is only logarithmically higher.

**Proposition K.2.** *The mapping* $\mathbf{b} \mapsto \mathbf{B}_{\mathbf{F}_{opt}}\mathbf{b}$*, where* $\mathbf{b} \in \mathbb{R}^n$*, may be evaluated in* $\mathcal{O}(n \log^2(n))$ *time.*

*Proof.* First, notice that the matrix $\mathbf{C_F}$ is one-to-one; indeed, this is immediately implied by the factorization $\mathbf{S} = \mathbf{B_F} \mathbf{C_F}$. By Theorem 1.2.1 (P6) of (Campbell & Meyer, 1979), any one-to-one matrix $\mathbf{T}$ admits the following representation for its pseudoinverse:

$$\mathbf{T}^\dagger = (\mathbf{T}^* \mathbf{T})^{-1} \mathbf{T}^*.$$

We compute:

$$
\begin{aligned}
\mathbf{C}_{\mathbf{F}}^\dagger &= \left(\mathbf{C}_{\mathbf{F}}^* \mathbf{C}_{\mathbf{F}}\right)^{-1} \mathbf{C}_{\mathbf{F}}^* \\
&= \left(\left(\mathbf{F}^* \sqrt{\Sigma} \mathbf{F} \mathbf{E}\right)^* \mathbf{F}^* \sqrt{\Sigma} \mathbf{F} \mathbf{E}\right)^{-1} \left(\mathbf{F}^* \sqrt{\Sigma} \mathbf{F} \mathbf{E}\right)^* \\
&= \left(\mathbf{P} \mathbf{F}^* \sqrt{\Sigma}^* \mathbf{F} \mathbf{F}^* \sqrt{\Sigma} \mathbf{F} \mathbf{E}\right)^{-1} \left(\mathbf{F}^* \sqrt{\Sigma} \mathbf{F} \mathbf{E}\right)^* \\
&= \left(\mathbf{P} \mathbf{F}^* |\Sigma| \mathbf{F} \mathbf{E}\right)^{-1} \left(\mathbf{F}^* \sqrt{\Sigma} \mathbf{F} \mathbf{E}\right)^*
\end{aligned}
$$

Now, the matrix $\mathbf{P}\mathbf{F}^*|\Sigma|\mathbf{F}\mathbf{E}$ is Toeplitz, since $\mathbf{F}^*|\Sigma|\mathbf{F}$ is circulant, and $\mathbf{P}$, $\mathbf{E}$ combine to select out the top-left $n \times n$ square of $\mathbf{F}^*|\Sigma|\mathbf{F}$. Notice that $\mathbf{P}\mathbf{F}^*|\Sigma|\mathbf{F}\mathbf{E}$ is not circulant, and cannot therefore be diagonalized by the $n$-dimensional Fourier transform.

The development of Section 4 yield the representation:

$$\mathbf{S} = \mathbf{P}\mathbf{F}^*\Sigma\mathbf{F}\mathbf{E},$$

which implies that matrix-vector products with the matrix $\mathbf{S}$ may be computed in $n \log n$ time by the use of the FFT. Similarly, matrix-vector products with $\left(\mathbf{F}^* \sqrt{\Sigma} \mathbf{F} \mathbf{E}\right)^*$ may be computed in $n \log n$ time.

Therefore the computational cost of computing the mapping $\mathbf{b} \mapsto \mathbf{S}\mathbf{C}_{\mathbf{F}}^\dagger \mathbf{b}$ can be upper bounded by the maximum of $n \log n$ and the cost of computing the mapping $\mathbf{v} \mapsto (\mathbf{P}\mathbf{F}^*|\Sigma|\mathbf{F}\mathbf{E})^{-1}\mathbf{v}$.

The cost of computing this mapping is, in turn, bounded by the cost of inverting a general (full-rank) Toeplitz system, since $(\mathbf{P}\mathbf{F}^*|\Sigma|\mathbf{F}\mathbf{E})^{-1}\mathbf{v}$ may be alternatively characterized as the solution $\mathbf{x}$ to the equation $\mathbf{P}\mathbf{F}^*|\Sigma|\mathbf{F}\mathbf{E}\mathbf{x} = \mathbf{v}$. The computational cost of solving such a system is known to be $n \log^2(n)$; see, e.g., (de Hoog, 1987). $\square$

**Lemma K.1.** *For a real-valued vector* $\mathbf{v}$*, let* $\hat{\mathbf{v}}$ *represent its discrete Fourier transform. If* $\hat{\mathbf{v}}$ *as no purely real, negative entries, then letting* $\sqrt{\cdot}$ *denote the (pointwise) principal branch of the square root and* $\mathbf{F}$ *the matrix representation of the Fourier transform, the matrix* $\mathbf{F}^* \sqrt{\hat{\mathbf{v}}} \mathbf{F}$ *is real-valued.*

*Proof.* Conjugate symmetry of the DFT states that for a $j$-dimensional real-valued vector $\mathbf{x}$, $\hat{\mathbf{x}}[m] = \overline{\hat{\mathbf{x}}}[j-m]$, and that the converse also holds–that if $\hat{\mathbf{x}}$ has this symmetry, $\mathbf{x}$ is real-valued. This can be seen by examining the action of conjugation of $\mathbf{x}$ on the Fourier transform $\hat{\mathbf{x}}$.

Now, by the assumptions on $\hat{\mathbf{v}}$ and the choice of the principal branch of the square root[5], if $\hat{\mathbf{v}}$ has this conjugate symmetry, so does $\sqrt{\hat{\mathbf{v}}}$. Therefore there is some real-valued vector $\mathbf{y}$ such that $\hat{\mathbf{y}} = \sqrt{\hat{\mathbf{v}}}$. The matrix $\mathbf{F}^* \sqrt{\hat{\mathbf{v}}} \mathbf{F}$ represents convolution with $\mathbf{y}$ in the standard basis, and hence is real-valued. $\square$

---

[5]These assumptions can be avoided, though at the cost of taking care in choosing the square root of the negative elements of $\hat{\mathbf{v}}$ to preserve the appropriate symmetry.

**Remark.** Lemma K.1 can be understood as a statement about the solvability of a certain repeated-convolution equation over real-valued functions (the equation $g * g = f$). We suspect that this fact has been observed in the harmonic analysis literature as a general property of all Fourier transforms; we could find no reference. The symmetries discussed above take a slightly different form in the continuous and noncompact case (IE, Fourier transform on real-valued function on $\mathbb{R}^d$) and the finite-dimensional Fourier transform here, so we choose to prove this statement in this limited setting.

