# OpenReview forum: "Multi-Epoch Matrix Factorization Mechanisms for Private Machine Learning"
_ICLR.cc/2023/Conference — Submitted to ICLR 2023_

### Official Review · Reviewer_rY9u · 2022-10-23

**Confidence:** 4
**Correctness:** 4
**Technical Novelty And Significance:** 3
**Empirical Novelty And Significance:** 3
**Recommendation:** 5

**Clarity, Quality, Novelty And Reproducibility:**

I think the main part of the paper is clear. But I think the novelty is less and it seems there are not so many technical approaches.

**Strength And Weaknesses:**

Strength:
The authors prove a new theorem bounding sensitivity for multi-dimensional data contributions based on an extension of the Birkhoff-von Neumann theorem. It allows to reduce the problem to that of measuring sensitivity for scalar contributions alone.

They show that the algorithms proposed in Denisov et al. (2022) can be extended in present setting.

To reduce the cost, they propose and investigate an approach based on the Fast Fourier which is near optimal for the single-pass setting and can be efficiently extended to handle multiple passes.

Weakness:
The techniques applied in the present paper based on the methods in Denisov et al. (2022). The authors extend it to the multi-dimensional and multi-participation setting. I think the novelty of the paper is less.




**Summary Of The Paper:**

In this paper, the authors provide new differentially private (DP) mechanisms for gradient-based machine learning (ML) training involving multiple passes (epochs) of a dataset, substantially improving the achievable privacy-utility-computation tradeoff. They propose a framework for computing the sensitivity of matrix mechanisms under general participation patterns. They extend the results of Denisov et al. (2022) to the optimization problems corresponding to these generalized notions of sensitivity. They propose and analyze a computationally-efficient factorization based on the Fourier transform.



**Summary Of The Review:**

In this paper, the authors introduce DP mechanisms for gradient-based machine learning training involving multiple passes of a dataset, substantially improving the achievable privacy-utility-computation tradeoffs. The key contribution is an extension of the online matrix factorization DP mechanism to multiple participations. I think the novelty of the paper is little, however they significantly improves the privacy-utility tradeoffs while maintaining practically feasible computation costs.

---

> ### Author Response · Authors · 2022-11-09
> **Theoretical Results are Non-Trivial.**
>
> Thank you for your review.
>
> We would like to highlight that our theoretical contributions are non-trivial. We provide three such contributions: 1) extending sensitivity analysis to the multi-participation case, 2) proving optimal solutions exist when optimizing for these multi-participation sensitivities, and 3) proving a more computationally efficient FFT algorithm. We discuss our theoretical contributions more at the end of the response.
>
> **Theoretical challenges and novelty:** On the hardness of the generalization to multiple participations: While we have intentionally kept the statements of our theorems and results as symmetric with the existing single-participation literature, the mathematical details of the problem are in fact substantially different. For example, the single-participation analogue of Cor 2.1 is a (literal) one-liner, whereas Thm H.1 requires significant technical difficulty.  Another indicator of a "phase change" in the difficulty of the problem is the fact that multiple participations in general leads to exponentially many constraints in the optimization problem (and correspondingly, that even computing sensitivity becomes NP-hard even for d=1 in general). Hence, a major contribution was identifying (k, b)-participation as a both tractable, realistic, and highly effective participation schema.
>
> **Empirical Contributions:** Moving to our empirical contributions, we show significant benefits in the privacy-utility tradeoffs for all our algorithms. Considering our optimal matrices, we find that we outperform all prior work by ~5 percentage points across many privacy budgets.

---

> ### Author Response · Authors · 2022-11-17
> **Pending Questions?**
>
> Thank you for your review.
>
> If our response left any of your concerns unaddressed, please let us know.

---

### Official Review · Reviewer_SJLq · 2022-10-24

**Confidence:** 3
**Correctness:** 4
**Technical Novelty And Significance:** 2
**Empirical Novelty And Significance:** 2
**Recommendation:** 6

**Clarity, Quality, Novelty And Reproducibility:**

Regarding the clarity, I have provided a couple of pointers on improving the presentation of this paper, but the rest of it seems alright to follow.

The results are novel in the sense that prior work was generalised here with new analyses and that a computationally-efficient analogue was provided for the same. That said, I do have concerns about the quality, which I have explained in the previous section.

**Strength And Weaknesses:**

Strengths:
1. The analyses of the sensitivity of the matrix mechanisms looks quite interesting and involved to me. I like the idea of incorporating these generalised cases of multiple contributions by a single user (user-level DP) because that is a much more real-world case.
2. The reduction to the scalar case also looks cute! It appears to give a cleaner way of visualising what the sensitivity could be, and anticipating the amount of noise that would be needed for privacy.

Weaknesses:
1. My first complaint is not too severe, but it is about the writing of this paper. I think the writing looks a bit rushed and doesn't really give the best intuition of what or why something could be expected, or in other words, there are places where the technical discussion is not necessarily supported by enough "English text".I see that happening in Sections 3 and 4, mainly. It is a little hard to see in the theorems what certain expressions mean and what they entail. In Section 5, there isn't any such issue, but Figure 2 that lies on page 8, is decoded and explained at the end of page 9, which is a bit confusing and distracting.
2. I'm not completely convinced by the empirical results in the main body of this paper. Here's my issue. The optimal (but possibly computationally-inefficient) algorithm seems to work really well compared to all other prior work and this paper's FFT-based mechanism, but it is not really practical when $n$ is large because of its high runtime. I think the point of the authors here might be to show the promise of their new generalisation, but right now, it is not exactly useful in practice.
3. The more concerning issue for me is that the practical analogue (that is based on FFT) of their mechanism is not super accurate. In Figure 1, it doesn't look that impressive even for the more useful values of $\varepsilon$ ($\leq 10$). It seems to be quite close to the Kairouz'21 algorithm, except when $\varepsilon$ is between $5$ and $7$. The optimal (but slower) algorithms does notably better in this regime though, but that doesn't seem impressive because that algorithm is not really practical in real-world applications. For the multiple epoch setting, it doesn't seem that useful either. Therefore, if this is one if the highlights of this paper, I'm a bit disappointed, to be honest.
4. I don't understand why the authors chose to compare in regimes where $\varepsilon$ is more than $10$ or too high to provide any useful privacy guarantee. We do expect most non-trivial DP algorithms to become accurate in that range anyway, so what is the point exactly?
5. I wish the authors had plots in the paper comparing their FFT-based approach more with other prior work. I know they have plots in the appendix, but they're hard to read because the plots for the prior work are separate from those for this paper's algorithm.
6. I understand that the focus of this work is mainly towards practical DP ML via tools like privatised versions of SGD, but I wish that there were more about other applications, as well. Maybe, that would have asked for a separate paper, so I'm not *that* concerned about this. Would appreciate the authors' input on this though.

**Summary Of The Paper:**

This paper provides new differentially private (DP) mechanisms for gradient-based machine learning training that involves multiple epochs of a dataset, and improves on the prior results. The algorithms are based on extensions of the matrix factorisation DP mechanism for different settings, such as user-level DP and multiple epoch regime. As listed in the introductions, there are four main contributions of this paper.
1. It provides a new framework for analysing the sensitivity of the matrix mechanisms under a certain user-level DP setting. This involves proving a new theorem for high-dimensional contributions based on an extension of the Birkhoff-von Neumann theorem, allowing to reduce to a scalar problem, instead.
2. It extends the results of Denisov' 22 to the optimisation problems corresponding to generalised notions of sensitivity.
3. It also gives a computationally-efficient factorisation mechanism that is based on the Fourier transform, that it claims is near-optimal for the single-pass setting, and could be extended to multiple epochs.
4. It also provides empirical evaluations and comparisons of their algorithms with prior matrix mechanism and DP-SGD, and claims to outperform these methods.

**Summary Of The Review:**

For me, the weaknesses outweigh the strengths a bit here. I am open to changing my evaluation later, but for now, I think a few improvements could help with the quality of this work.

Update: based on the responses, I have bumped the score.

---

> ### Author Response · Authors · 2022-11-09
> **Optimal Matrix Mechanisms are Practical and Privacy Budgets are Reasonable.**
>
> Thank you for your review.
>
> We have improved the writing and added computation benchmarking as noted in our group update. Further, we emphasize that our optimal mechanisms are computational feasible for many steps up to 10,000. This makes it well-suited to both federated settings (n<2000 typically) and central settings (e.g., CIFAR10 where we have n=2000). However, because it scales poorly in the number of steps, our FFT method shows promise for larger step settings. In this setting, the optimal FFT mechanism achieves very comparable (though still worse) privacy-utility tradeoffs compared with the optimal matrix mechanism, while scaling polylog in the number of steps. We discuss these tradeoffs more in section 4 where we also give empirical runtimes. Below are detailed responses to each point.
>
> 1. **Writing:** We have since improved the writing in the updated version and ensure that Fig 2 appears on the same page as the description of it.
>
> 2. **Feasibility of the optimal method:** There are two computation costs involved in using optimal matrix factorizations: 1) Finding the optimal factorization (termed optimization cost in the main-text), and 2) computing noise during training (termed noise generation cost). Importantly, 1) needs to only be done once, and then the resulting matrices can be re-used to train any number of models. Such matrices could be pre-computed and shared globally via a common rqepository, for example. Computing optimal factorizations up to n=8000 took less than 24 hours with minimal code optimization and using only CPUs; an optimized GPU implementation would likely reduce these costs even further. For 2), we found brute-force generation of noise to incur negligible overhead compared to the computation costs of compute gradients / model updates on larger batches. We believe user-level DP is particularly important. Our algorithms here extend the FedAvg algorithm, which takes multiple local steps on each user's dataset before averaging, meaning fewer (private) averaging steps are needed. For example, our StackOverflow experiments are representative of [production use of DP-FTRL](https://ai.googleblog.com/2022/02/federated-learning-with-formal.html) which used 2000 steps.  [Adaptive Federated Optimization](https://arxiv.org/pdf/2003.00295.pdf) presents a suite of six benchmark optimization tasks using variants of FedAvg, all of which converge in 1500 - 4000 steps (rounds), well within the range of practicality for matrix factorization.
>
> 3. **FFT practicality:** The FFT mechanisms are indeed practical in many settings. The updated results with proper tuning shows that it achieves excellent privacy-utility tradeoffs. It achieves around a 4 percentage point increase over the Kairouz et al. method consistently across tighter privacy levels, and around 2 percentage points at higher privacy levels.
>
> 4. **On large epsilons:** Moving DP to widespread use in real-world ML is one of our primary motivations, and we believe using any model with a finite epsilon is a significant step in the right direction over non-DP training. Thus, it is useful to fully characterize privacy-utility tradeoffs up to a point of minimal accuracy loss. Observing Fig 1 and Fig 3, even for high privacy budgets, achieving high accuracy is a non-trivial task. We can see this because using DP-SGD significantly degrades  performance compared with the non-private baseline. Our methods outperform all others by significant margins. Here we get 3+ percentage points with prior work and over 7+ with DPSGD. In the more interesting regime of $\varepsilon < 10$, we still observe similar improvements. We remark that our $\varepsilon < 10$ regime corresponds to a $\leq1$-zCDP guarantee. This is stronger than traditional (\epsilon,\delta)-DP guarantees, is better than the standard set by US Census (\rho\approx 2.6), and is comparable to that of the production deployment by Google (\rho=0.81). For more detailed discussion on this, see [the Google AI blog post](https://ai.googleblog.com/2022/02/federated-learning-with-formal.html).
> In addition to privacy and utility, computation and dataset size are fundamental to the DP tradeoff space. Previous work shows model accuracy is relatively stable as batch sizes (or clients/round) are scaled together with noise to maintain a constant signal-to-noise ratio, which allows us to extrapolate to more realistic settings. For example the epsilon=18 stackoverflow results indicate epsilon=2.2 is possible by simply scaling the noise, dataset size and clients/round up by a factor of 6 (which is realistic in practice, e.g. [DP-FTRL in Google AI Blog Post](https://ai.googleblog.com/2022/02/federated-learning-with-formal.html) uses 6500 clients per round compared to 1000 in our experiments, and the 340,000 clients of StackOverflow is small for a production dataset).
>
> Responses to 5. and 6. are included below.

---

> > ### Author Response · Authors · 2022-11-09
> > **Continued... Our Work is Applicable to e.g., Mean Estimation.**
> >
> > 5. **Comparing FFT Empirically:** We are happy to include any suggested plots comparing FFT with prior work. We note that our main Fig 1 includes a full privacy-utility takeoff of the best FFT factorization compared with prior work, where `best’ is obtained via choosing the stamping by losses (which is computationally simple to compute) and then tuning the mechanism’s learning rate and momentum values as described in Appendix E.
> >
> > 6. **Application of our work beyond ML:** Indeed, our mechanisms operate broadly on settings that can be derived from the batch release of DP linear operators. The loss definition is e.g. applicable to DP mean estimation in this setting, which we have added some text for in Section 3. The results can be viewed in Appendix C.6 (Table 1). Because (and as noted in the bottom of the first paragraph of Section 5.1) these well align with the ordering of accuracy (the metric in our main application), we only show the results in terms of accuracy in the main text. We have added some description on this when we introduce the loss in Section 3.

---

> ### Author Response · Authors · 2022-11-17
> **Pending Questions?**
>
> Thank you for your review.
>
> If our response left any of your concerns unaddressed, please let us know.

---

### Official Review · Reviewer_6QXF · 2022-10-27

**Confidence:** 3
**Correctness:** 4
**Technical Novelty And Significance:** 3
**Empirical Novelty And Significance:** 3
**Recommendation:** 6

**Clarity, Quality, Novelty And Reproducibility:**

The article is well-organized and easy to follow, and the experimental results are also intuitive and clear.

The method proposed in this paper is relatively novel, especially the extension of the online matrix factorization DP mechanism to multiple participations and a Fourier-transform-based optimization.

There are little sentences with grammatical errors.


**Strength And Weaknesses:**

a)  This paper emphasises their contributions/novelties on existing differentially private(DP) mechanisms. The extension is reasoned and the solution method is proved, and the effectiveness of method is proved by extensive experiments on example-level DP and user-level DP.

b)  Author applies differentially private (DP) mechanisms with matrix factorization to gradient-based machine learning, and promot the achievable privacy-utility-computation tradeoffs.

c)  The formula proof of paper is rigorous and orderly, and the diagram of the experiment is also very intuitive.

d)  In section 4,the time complexity is analyzed theoretically. Could the authors report the running time of the proposed algorithm in the experiment? In this way, we can intuitively justify that Discrete Fourier Transform helps reduce computation.

e)  The transition from the background to the formula should be elaborated in more detail so that readers can understand more quickly.

f)  In the conclusions part, the following can be added. For example, weaknesses of work and future work discussion.


**Summary Of The Paper:**

This paper proposes a novel differentially private mechanism with multiple participations. Specifically, the author simplify per-iteration
vector contributions problem by using a sensitivity computing framework and rigorous proof is given. Under the problem reduction, a convex program for the construction of optimal matrix mechanisms is proposed, and a closed form solution for it is also provided. To further reduce computational consumption, a Fourier-transform-based mechanism is designed. In experimental part, full comparisons with prior matrix mechanism approaches and DPSGD demonstrates the effectiveness of the proposed method.


**Summary Of The Review:**

See "Strength And Weaknesses"

---

> ### Author Response · Authors · 2022-11-09
> **Empirical runtimes show feasibility for around 10,000 steps, which has practical use.**
>
> Thank you for your detailed reviews and appreciation of our algorithm’s benefits.
>
> We have added runtimes for our experiments as detailed below in d). We address weaknesses using the same lettering scheme as your review.
>
> d) **Empirical runtimes:** As noted in the group update, we have discussed the runtime for our experiments in the FFT Section 4 and in a new Appendix C. We note that the major challenge is optimizing matrices, and doing so past 10,00 steps (recall our setup has 2,000 steps) can become intractable quickly. However, our Optimal FFT decoder does not have any optimization costs and recovers much of the privacy-utility tradeoff present in the optimal mechanism. As for noise generation, all mechanisms, even those that scale as $O(n^2)$ are feasible for many steps.
>
> e) **Writing:** We have improved our writing and hope that it addresses your concern. If not, we are happy to make additional changes at your request. Not sure.
>
> f) **Discussion and conclusion:** Due to space constraints, we have unfortunately had to remove our discussion from the main text. We have placed it in Appendix G and included your suggested discussion.

---

> ### Author Response · Authors · 2022-11-17
> **Pending Questions?**
>
> Thank you for your review.
>
> If our response left any of your concerns unaddressed, please let us know.

---

### Official Review · Reviewer_PQTm · 2022-10-27

**Confidence:** 3
**Correctness:** 3
**Technical Novelty And Significance:** 2
**Empirical Novelty And Significance:** 3
**Recommendation:** 5

**Clarity, Quality, Novelty And Reproducibility:**

The paper is mostly-well written but some parts have missing important details or are relegated to prior works. The extension from single participation does not seem particularly challenging and quality mainly hinges on observed empirical improvements.

**Strength And Weaknesses:**

Strengths

1. The prefix-sum view of SGD and using matrix factorization techniques to privately release these have been instrumental in improving the utility in private machine learning tasks, especially in large-scale practical settings. Therefore, the problem that the authors study is important and the improvements can have immediate real-world impact in how private machine learning models are being trained.

2. The paper is mostly written-well; it gradually formulates the problem, and builds towards the key ideas and proposed approach.

3. The experimental results showcase improvement over prior approaches; for example they get a 5\% improvement over DP-SGD with privacy amplification with sub-sampling.

Weakness

1.  The section on FFT seems half baked and not so related with the overall picture. The main goal in the paper to demonstrate empirical improvement but it seems to be that empirical results presented in the main text don't even use this method; correct me if wrong.


2. $(k,b)$ participation - I did not fully understand the choice and use of assumption that a point participates once every $b$ iterations. It seems to me that it is only used to argue that $|\Pi|=b$; in that case, why not simply explicitly assume this? If not, where is the additional structure useful?

3. At parts, I felt that some details are missing in order to completely understand the arguments, or the authors expect the reader to be familiar with Denisov et al. (2022). For example, in Corollary 3.1, the optimal dual solution computed, but how to get the primal solution from here? It also is the case with some experimental details (see below). It would be helpful if the authors can make the content more self-sufficient.

4. Experiments: I do not understand the description of algorithms MF1,6e and MF6,6e; for MF1,6e the authors say that they use Theorem 2.1 to use "the single participation optimized matrix", but Theorem 2.1 is about reducing $d$ dimensional instances to scalar instances; what am I missing here? About MF6,6e, it says, is "our approach .." -- which theorem/procedure description should I be looking at for this?

**Summary Of The Paper:**

The paper studies the problem of privately releasing prefix sums in the adaptive continual release model via matrix factorization techniques. Their main results are 1.extension of prior work which was limited to single participation to general participation patterns and 2. a more efficient method based on fast Fourier transform. The authors conduct extensive experiments which show promising results over prior approaches.


**Summary Of The Review:**

I think the topic of the paper is timely and the authors make interesting contributions with promising empirical results. The downsides are missing/unclear details as well as lack of coherence among parts of the paper, like the one based on FFT.

---

> ### Author Response · Authors · 2022-11-09
> **Participation Assumption is Critical to Theoretical Analysis.**
>
> Thank you for your agreement on the importance of the topic and reviews.
>
> We have made improvements to our presentation that we believe address your concerns there. We have also improved our description of the different algorithms for experiments.
>
> Detailed responses to the weaknesses and questions below, using your same numbering.
>
> 1. **Writing of FFT:** As in our group update, we have improved the writing of the FFT method. We remark that our main results do leverage the FFT mechanism in Figure 1, and that it shows significant improvement over mechanisms with comparable compute. For example, the FFT Optimal Decoder outperforms DPSGD with amplification to $\varepsilon \approx 4$ whereas the Online Honaker of Kairouz et al. can only go as low as $\varepsilon \approx 8$, even after our improvements to it.
>
> 2. **Participation Assumption:** We assume (k,b) participation for three main reasons: 1) it enables tight sensitivity analysis (via our Section 2 methods, now clarified explicitly via Eq. (3)); 2) precisely controlling participation via an exact separation of b leads to a larger feasible set of Eqs. (5) and (6). This is in fact a core contribution of the paper, which we will make more clear: we are able to produce better privacy/accuracy tradeoffs because the mechanisms are optimized for a very specific participation pattern; see the newly included Figure 2, which highlights the precise structure in the computed optimal matrices.  Finally, 3) (k, b)-participation closely matches the standard ML training pipeline. Indeed, this only requires that global shuffling is performed instead of shuffling per epoch, i.e., one shuffle is performed before training. Our updated manuscript describes this in the second-to-last paragraph of page 3.
>
> 3. **Technical Writing:** Unfortunately, due to space limitations, we were unable to include all details in the main text. Our updated manuscript includes a more self-contained view. We include more links in the main-text to the necessary appendices.
>
> 4. **Empirical Mechanism Descriptions:** Apologies, our original manuscript linked to the wrong Theorem. Our updated version now properly links to our updated Corollary 2.1. In more detail, MF(1,6e) is a matrix optimized for 1 epoch participations (i.e., the work of Denisov et al.) but, using our Section 2 and Corollary 2.1 to apply this matrix to a setting of >1 epochs. Our work, MF(6,6e) corresponds to directly optimizing a matrix for the 6 epoch geometry, which requires usage of our Section 2 to do. We have better explained this in the main-text.
>
> **Theoretical challenges and novelty:** On the hardness of the generalization to multiple participations: While we have intentionally kept the statements of our theorems and results as symmetric with the existing single-participation literature, the mathematical details of the problem are in fact substantially different. For example, the single-participation analogue of Cor 2.1 is a (literal) one-liner, whereas Thm H.1 requires significant technical difficulty.  Another indicator of a "phase change" in the difficulty of the problem is the fact that multiple participations in general leads to exponentially many constraints in the optimization problem (and correspondingly, that even computing sensitivity becomes NP-hard even for d=1 in general). Hence, a major contribution was identifying (k, b)-participation as a both tractable, realistic, and highly effective participation schema.

---

> ### Author Response · Authors · 2022-11-17
> **Pending Questions?**
>
> Thank you for your review.
>
> If our response left any of your concerns unaddressed, please let us know.

---

### Author Response · Authors · 2022-11-09
**Group Update: Improved Version Updated**

Thank you for your reviews.

First, we would like to highlight the empirical benefits of our generalized approach. Observing our updated Fig 1 (changes explained in more detail below), we see two major improvements of our work: 1) there is significant (about 15 percentage points) compared with what is possible using only Denisov et al., and 2) for the tighter epsilon regimes ($\varepsilon < 10$) there is a 3+ percentage point gain compared to any prior work (or 5+ to DPSGD with amplification).

**Writing and self-contained:** second, we have updated our writing which we believe addresses concerns surrounding clarity. We have focused all our sensitivity analysis to Section 2 which we now use to prove the sensitivity of our FFT mechanism. We improved the discussion surrounding the importance of the FFT mechanism versus the optimal factorizations, including on computation. We have made appendices self-contained and make fewer leaps in the main-text.

**Theorem 2.1:** Finally, we noticed that our Theorem 2.1 did not hold in general for all C^TC matrices. We provide a counterexample showing this in Appendix F.2. This gap makes the challenge of proving sensitivity a larger challenge. We fixed and updated our sensitivity analysis accordingly finding that under certain mild constraints the sensitivity can be efficiently bounded. We find these cases align well with practice and that all our mechanisms satisfy these conditions. These changes did not impact any conclusions or empirical results, though they do highlight subtleties arising from passing to the multi-epoch setting.

Third, we have improved our empirical evaluation in five major ways:
1. **Comparing to Denisov et al.:** we have altered the main results in Figure 1 to properly differentiate our contributions versus prior work. Originally, we had attributed the MF(k=1,n=2000) mechanism to Denisov et al.. This is not accurate because it requires our Section 2 analysis to bound the sensitivity so that it can be used in this $k=20$ setting. We have run a mechanism, MF(k=1,n=100) which corresponds to 1 epoch (their limitation) and attributed that to Denisov, as it should be. To show the benefits of optimizing for the target geometry, we use the optimization algorithm of Denisov to generate matrices optimal for 1 epoch, but use our section 2 analysis (and the ‘stamping’ construction of Section 5) to apply them to our setting; we therefore attribute the resulting MF(k=1,n=1000)x2 mechanism to our work. We observe our optimal mechanisms, which include our Section 3 analysis to directly optimize for the 20 epoch geometry, significantly outperform all others.

2. **Honaker/Kairouz et al.:** we properly note that we actually make improvements on the Kairouz et al., mechanism by zeroing out the noise for virtual steps used to “complete the tree”. We describe this in Appendix C.3. This led to some improvements on their work, which we still attribute to their mechanism in our results.

3. **FFT:** we separate out two separate factorizations based on the FFT mechanism. Our original results were based on what we have renamed to be the FFT Optimal Decoder, because its running time is actually slightly larger ($O(ndlog^2n)$) than using the exact FFT decoder from Algorithm 1. We have added results for the mechanism corresponding exactly to the FFT decoder to the main figure as well.

4. **Hyperparameter tuning:** we re-ran our experiments by tuning the momentum values for each mechanism. We had assumed initially that each would use similar values (i.e., we only tuned for prefix sum and used that for the rest). Now we have tuned them all independently and found much smoother results, as shown in Fig 1.

5. **Computational benchmarking:** we added benchmarking for computational analysis to clearly delineate that the main bottleneck is generating matrices, and that generating noise for all mechanisms is typically feasible.

---

### Author Response · Authors · 2022-11-12
**Pending Questions?**

Thank you all for your reviews. We greatly appreciate your feedback.

If our responses leave any of your questions unanswered or concerns unaddressed, please let us know.

---

### Decision · Program_Chairs · 2023-01-20

**Decision:**

Reject

**Justification For Why Not Higher Score:**

I think the paper was borderline after the major changes in the manuscript, and slightly below borderline beforehand. The reviewers thought the changes were quite substantial and didn't think basing the review on the quite new version of the manuscript would be fair, and thought instead that the paper would benefit from an extra round of reviewing.

**Justification For Why Not Lower Score:**

N/A

**Metareview: Summary, Strengths And Weaknesses:**

The reviewers appreciated the changes in the revision the authors made. However, it was thought by several reviewers that the changes in the revision were quite major and substantial, in particular, major rewrites of sections, additional conditions in Corollary 2.1 and new sensitivity analysis, and it was thought that the paper would benefit from a new round of reviewing given such changes. If evaluated prior to the technical changes, the enthusiasm was a bit weaker.